# CLR01 protects dopaminergic neurons in vitro and in mouse models of Parkinson's disease

Nora Bengoa-Vergniory [1], Emilie Faggiani[2,3], Paula Ramos-Gonzalez[4], Ecem Kirkiz[1], Natalie Connor-Robson [1], Liam V. Brown[5], Ibrar Siddique[6], Zizheng Li[6], Siv Vingill[1], Milena Cioroch[1], Fabio Cavaliere[4], Sarah Threlfell [1], Bradley Roberts [1], Thomas Schrader[7], Frank-Gerrit Klärner[7], Stephanie Cragg [1], Benjamin Dehay [2,3], Gal Bitan [6], Carlos Matute[4], Erwan Bezard [2,3] & Richard Wade-Martins [1]✉

Parkinson's disease (PD) affects millions of patients worldwide and is characterized by alpha-synuclein aggregation in dopamine neurons. Molecular tweezers have shown high potential as anti-aggregation agents targeting positively charged residues of proteins undergoing amyloidogenic processes. Here we report that the molecular tweezer CLR01 decreased aggregation and toxicity in induced pluripotent stem cell-derived dopaminergic cultures treated with PD brain protein extracts. In microfluidic devices CLR01 reduced alpha-synuclein aggregation in cell somas when axonal terminals were exposed to alpha-synuclein oligomers. We then tested CLR01 in vivo in a humanized alpha-synuclein overexpressing mouse model; mice treated at 12 months of age when motor defects are mild exhibited an improvement in motor defects and a decreased oligomeric alpha-synuclein burden. Finally, CLR01 reduced alpha-synuclein-associated pathology in mice injected with alpha-synuclein aggregates into the striatum or substantia nigra. Taken together, these results highlight CLR01 as a disease-modifying therapy for PD and support further clinical investigation.

[1] Oxford Parkinson's Disease Center (OPDC) and Department of Physiology, Anatomy and Genetics, Oxford University, South Parks Road, Oxford OX1 3QX, UK. [2] Institut des Maladies Neurodégénératives, UMR 5293, Univ. de Bordeaux, F-33000 Bordeaux, France. [3] CNRS, Institut des Maladies Neurodégénératives, UMR 5293, F-33000 Bordeaux, France. [4] Departamento de Neurociencias, Achucarro Basque Center for Neuroscience, and Centro de Investigación Biomédica en Red en Enfermedades Neurodegenerativas (CIBERNED), Universidad del País Vasco (UPV/EHU), S-48940 Leioa, Spain. [5] Mathematical Institute, Oxford University, Oxford OX2 6GG, UK. [6] Department of Neurology, Brain Research Institute and Molecular Biology Institute, University of California, Los Angeles, 635 Charles E Young Drive South, Gordon 451, Los Angeles, CA 90095, USA. [7] Institute of Organic Chemistry, University of Duisburg-Essen, Essen 45117, Germany. ✉email: richard.wade-martins@dpag.ox.ac.uk

Parkinson's disease (PD) is the second most common neurodegenerative disorder affecting millions of people worldwide presenting with resting tremor, altered gait, bradykinesia, rigidity, hyposmia, constipation, and rapid eye movement sleep behavior disorder[1]. Classic neuropathological examination of PD patient post-mortem tissue shows a loss of dopaminergic neurons in the substantia nigra (SN) pars compacta (SNc) of the midbrain, as well as an accumulation of α-synuclein (α-syn) in inclusions known as Lewy bodies (LBs) and Lewy neurites[2]. It was previously thought that fibrils were the only pathogenic α-syn species within LBs in the midbrain of PD patients, but several lines of research have shown that α-syn oligomers are also important in PD pathogenesis[3]. Molecular tweezers have shown potential for tackling pathological protein aggregation[4,5]. These compounds inhibit key interactions in the self-assembly of amyloidogenic protein by binding to positively charged amino acid residues and disrupting both hydrophobic and electrostatic interactions. Specifically, the molecular tweezer CLR01 binds to lysine residues crucial for α-syn oligomerization and aggregation[6]. CLR01 has been shown to inhibit α-syn aggregation in cell lines and zebra fish[7], and, interestingly, to be neuroprotective in a triple transgenic Alzheimer's disease mouse model[8]. There is still a lack of thorough translational studies of neurodegenerative therapeutic approaches combining state-of-the-art models, such as induced pluripotent stem cell (iPSC)-derived dopaminergic cultures, humanized transgenic animal models, and animals inoculated with PD brain extracts.

iPSC-derived dopaminergic cultures have been shown to recapitulate midbrain development, and neurons survive in vivo and integrate into Parkinsonian cellular networks[9]. Using iPSC-derived dopaminergic cultures, we have detected cellular disease phenotypes that recapitulate findings from post-mortem PD cases[10,11]. We have also shown previously that expressing the complete human SNCA locus as a transgene in mice on a $Snca^{-/-}$ background recapitulates the spatial and temporal expression of the endogenous human α-syn[12]. Transgenic animals exhibit age-dependent motor behavioral phenotypes, dopaminergic cell loss, as well as deficits in dopamine neurotransmission. We have also previously demonstrated that LB-containing fractions of Parkinsonian brains can be isolated, and that injection of this purified material into the SN of wild-type (WT) mice recapitulates Parkinsonian degeneration of dopaminergic neurons after 4 months with α-syn aggregation spreading into connected areas of the brain[13].

In this study, we show that CLR01 reduces both the aggregation of recombinant α-syn and dissociates pre-aggregated α-syn. We show that CLR01 reduces α-syn aggregation in LB extracts from PD patients in vitro. iPSC-derived dopaminergic cultures insulted with LB extracts show shortening and blebbing of their processes, which is rescued by CLR01 treatment. In microfluidic chambers, CLR01 also reduces α-syn aggregation and transport. Our in vivo results show that CLR01 rescues motor behavior defects and reduces α-syn aggregates and accumulation of α-syn oligomers in the SN of SNCA transgenic animals dosed at 12 months, but not at 6 months, and only partially at 18 months. These data indicate that the timing of treatment within a therapeutic time window is critical when considering CLR01 for use in the clinic. Finally, CLR01 reduces pathology in two different α-syn injection models, namely an LB-injection model and a pre-formed fibril injection model, demonstrating broad therapeutic potential.

## Results

### CLR01 impairs α-syn aggregation in vitro
CLR01 binds specifically with high on–off rate to lysines and, therefore, its specificity is not to a particular protein but to the process of abnormal protein self-assembly itself. CLR01 has been shown to be suitable for a high therapeutic dosing window in rodent models that tolerate high doses of this compound well[14]. These data highlight the need to investigate the anti-aggregation and neuroprotective effects of this molecule further, to determine whether it is promising as a disease-modifying treatment for PD. To test whether CLR01 was able to have an impact on recombinant α-syn aggregation in vitro, we performed an electron microscopy (EM) time-course of recombinant α-syn aggregation. Recombinant α-syn was shaken at 250 r.p.m. and 37 °C for 1, 2, 3, 4, 5, and 10 days in the presence or absence of CLR01. In the absence of CLR01, recombinant α-syn aggregated, forming both fibrils and amorphous structures representative of oligomers, whereas in the presence of CLR01 aggregation was severely impaired (Supplementary Fig. 1A). We then evaluated whether CLR01 could reduce the aggregation of recombinant α-syn pre-formed oligomers and fibrils. Recombinant α-syn species were incubated in the presence or absence of CLR01 for a further 10 days and then evaluated by EM. CLR01 reduced the presence of oligomers and fibrils in these preparations (Supplementary Fig. 1B), which is in agreement with the effects of CLR01 on recombinant α-syn species previously shown by Prabhudesai et al.[7]. We then tested whether CLR01 could reduce the presence of aggregates in LB extracts from PD patients, which were produced as previously described[13,15,16]. LB extracts and their corresponding non-LB (noLB) control extracts were subjected to the α-syn proximity ligation assay (AS-PLA) and EM to evaluate the presence of oligomers[17,18] and the morphology of the aggregates (Supplementary Fig. 2), respectively. We have previously shown that we can detect α-syn oligomers with AS-PLA, which α-syn oligomers accumulate in the human where LB pathology is barely detectable[17], and that oligomeric deposition can be modulated[18]. Here we found that α-syn oligomers could only be detected in untreated LB extracts but were no longer detected when treated with CLR01 for 10 days (Supplementary Fig. 2A–C). We could also only find structures resembling LBs in LB extracts by EM (Supplementary Fig. 2D), which were greatly reduced upon incubation with CLR01. In summary, CLR01 inhibited α-syn aggregation and caused dissociation of both oligomeric and fibrillar α-syn.

### α-Syn aggregation and toxicity are reduced by CLR01 in human dopaminergic cultures
We initially explored the effects of CLR01 and CLR03 (an inactive analog) in cell lines. We transfected neuroblastoma SH-SY5Y cells with a plasmid over-expressing α-syn and measured the resulting AS-PLA signal (Supplementary Fig. 3A). Cells expressing the plasmid accumulated high amounts of intracellular α-syn oligomers and were treated by the addition of CLR01 to evaluate its ability to penetrate cells and to determine an $EC_{50}$ for the compound. We found that CLR01 was able to efficiently dissociate oligomers and/or prevent their aggregation with an $EC_{50}$ of 85.4 nM (Supplementary Fig. 3B, C) but that CLR03 did not have a significant effect on the aggregation status of α-syn. To further explore the therapeutic potential of CLR01, we next tested its ability to reduce α-syn aggregation and toxicity in iPSC-derived dopaminergic neuron cultures. Treatment with LB or noLB extracts produced different surface textures upon microscopic inspection. Whereas we could observe clear neuronal cell cultures in control and noLB samples, LB-treated cells exhibited an abnormal texture indicating deposition of putative proteinaceous material on the cell surface (Fig. 1a and Supplementary Fig. 4A). However, when pretreated with CLR01, this material or texture disappeared. The number of dead cells labeled by To-pro was unchanged after incubation of

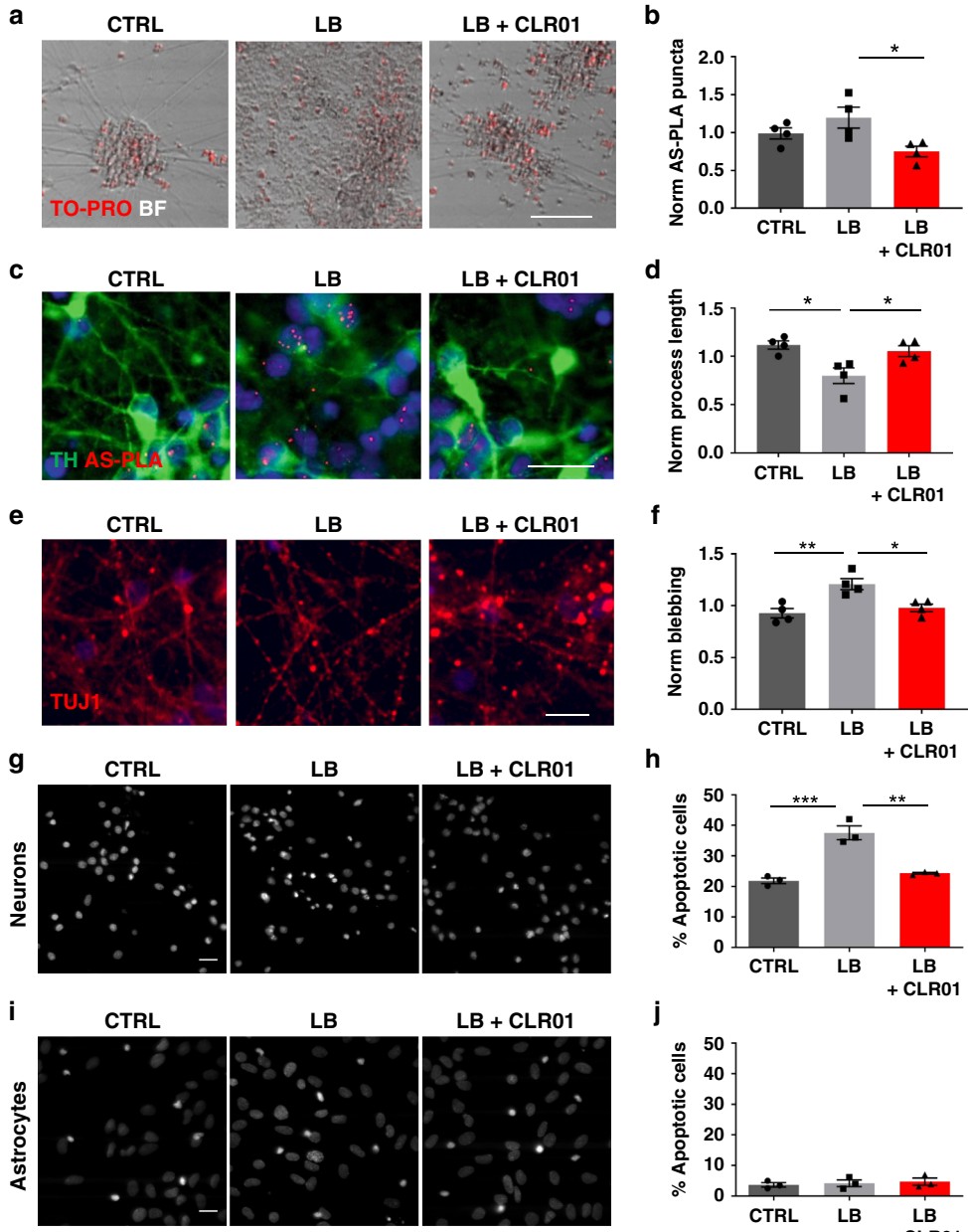

**Fig. 1 CLR01 reduces α-syn aggregation and toxicity in iPSC-derived dopaminergic and primary rat cortical cultures. a, c, e** Representative brightfield and immunofluorescent images of live and dead cells (labeled by To-pro), dopaminergic cells (TH-positive), AS-PLA (red puncta), and neuronal processes of four independent control cell lines analyzed. Cells were treated with LB extracts, which had been pretreated with CLR01 or PBS as a negative control. LB treatment caused an abnormal deposition of material in treated wells, accompanied by increased AS-PLA signal process degeneration, which were reduced by CLR01 pre-treatment. Scale bars = 100, 25, 25 μm, respectively. **b, d, f** Quantification of indicated parameters (from **c** and **e**) of four independent control cell lines analyzed using a one-way ANOVA (Sidak's). **b** $F_{(2, 9)} = 5.154$, $p = 0.0213$. **d** $F_{(2, 9)} = 7.309$, $p = 0.0112$, $p = 0.0356$. **f** $F_{(2, 9)} = 10.67$, $p = 0.0038$, $p = 0.0126$. **g, i** Representative images of neuronal (top) and astrocytic (bottom) primary rat cultures stained with DAPI ($n = 3$ independent cultures and treatments). Scale bars 20 μm. **h, j** Quantification of apoptotic cells (expressed as % from total) analyzed using a one-way ANOVA (Sidak's, $n = 3$ independent cultures and treatments). **k** $F_{(2, 6)} = 35.09$, $p = 0.0005$, $p = 0.0012$. *$p < 0.05$, **$p < 0.01$, ***$p < 0.001$. For all appropriate panels, data are presented as mean values ± SEM. BF brightfield, Norm normalized.

brain extract with CLR01 (Fig. 1a and Supplementary Fig. 4A, G, I). As a next step, we evaluated the levels of α-syn oligomers in the culture and the number of tyrosine hydroxylase (TH)+ cells (Fig. 1b–c and Supplementary Fig. 4B–C, H, J). The number of TH+ cells was unchanged but we detected an upregulation of AS-PLA puncta in LB-treated cells, which was reduced by 10 days of 10 μM CLR01 pre-treatment. Finally, as axonal degeneration has been shown to precede cell death in PD[19], we analyzed processes labeled by the Tuj1 antibody to βIII-tubulin (Fig. 1d–f and

Supplementary Fig. 4D–F). NoLB extracts did not affect process length or blebbing, whereas LB extract treatment shortened neuronal processes and increased blebbing, which was reduced by CLR01 pre-treatment back to control levels. This indicated that LB extract treatment caused axonal degeneration of iPSC-derived dopaminergic cultures, whereas CLR01 rescued the neurons from these toxic effects. Finally, we exposed iPSC-derived dopaminergic cultures to LB extracts pretreated with increasing concentrations of CLR01 or the inactive analog CLR03.

CLR01 showed a dose-dependent reduction of α-syn oligomers in dopaminergic cultures whereas CLR03 had no effect (Supplementary Fig. 5A, B).

To test whether CLR01 could reduce α-syn aggregation and toxicity in other neural populations, we treated primary rat neuronal and astrocytic cultures with LB extracts in the presence or absence of CLR01 (Fig. 1g–j). LB treatment induced cell death in the neuronal cultures, but not in astrocytes, in agreement with our previous findings[20], and these effects were prevented by CLR01 treatment. Altogether, these data suggest that CLR01 is able to reduce α-syn aggregation and toxicity in human iPSC-derived dopaminergic cultures and primary rat cultures.

**CLR01 prevents α-syn aggregation in microfluidic chambers.** To investigate how α-syn is taken up and transported, we used microfluidic chambers that allow axonal separation of two distinct cellular populations. As depicted in Fig. 2a in our first experiment, we plated cells on both sides, the home and the insult chamber, and kept the medium flow at all times towards the insult chamber. This flow system prevents any passive diffusion occurring from the insult chamber to the home chamber, therefore allowing us to determine that, if any insult is detected in the home chamber, it would have been transported along an axon (either by anterograde or retrograde transport). AS-PLA showed that when we insult iPSC-derived dopaminergic cultures with recombinant α-syn oligomers in the insult chamber, we can detect this insult in the home chamber, indicating that oligomers are actively transported along the axons of these cells (Fig. 2b). In the presence of CLR01, the level of AS-PLA signal is greatly reduced in the home chamber, indicating that CLR01 treatment protects neurons from an oligomeric insult. To test whether there was a specific involvement of retrograde transport in this process, we examined the same paradigm without seeding cells in the insult chamber (Fig. 2c). If an insult is detected in the home chamber in the absence of cell bodies in the insult chamber under the same flow conditions, the insult must have been transported by axonal retrograde transport. Upon recombinant α-syn oligomer insult at the terminals, we detected an increase in AS-PLA in the dopaminergic neuron cell bodies in our cultures (Fig. 2d), which indicated that oligomers are moved by retrograde transport. Application of CLR01 reduced the levels of AS-PLA seen after α-syn oligomer insult.

We therefore decided to study whether α-syn interacted with retrograde and anterograde transport proteins (dynein and kinesin, respectively) through PLA, and whether these interactions could be inhibited by CLR01. α-syn-dynein-PLA revealed that dynein does interact with α-syn, and that this interaction can be increased by insulting dopaminergic cultures with α-syn monomers, which is then reduced by CLR01 treatment (Supplementary Fig. 6A, B). When we examined the interactions between α-syn and kinesin through α-syn-kinesin-PLA we could detect a clear upregulation of the signal upon α-syn monomer insult, which was depleted with CLR01 treatment. This indicates that CLR01 is able to reduce this interaction and therefore influence α-syn transport (Supplementary Fig. 6C, D).

In summary, these data show that recombinant α-syn oligomers are actively transported along axons. Importantly, CLR01 reduces α-syn oligomeric pathology following active transport efficiently in iPSC-derived dopaminergic neurons and it is specifically able to reduce interaction of α-syn with axonal transport proteins.

**CLR01 reduces α-syn aggregation and its detrimental effects in vivo.** Our laboratory has previously described a human α-syn-overexpressing mouse model (SNCA-OVX), which showed

accurate spatial and temporal expression of human α-syn in a mouse Snca-null background. Overexpression of α-syn in this humanized model led to age-dependent neuronal loss, impaired dopamine release, altered vesicle clustering, and motor defects[12]. Here we conducted further analysis and found that, although SNCA-OVX mice do not show pronounced whole-brain gliosis, the SN has increased levels of GFAP (glial fibrillary acidic protein, astrocytes) and a shift in morphology for microglia (Iba1-positive cells) (Fig. 3a–d and Supplementary Fig. 7). Microglial cells in the SN are preferentially ramified (homeostatic) in WT animals, yet microglia of SNCA-OVX animals adopt a more ameboid morphology, which is suggestive of an activated state. We have reported previously that these animals do not exhibit LB-like aggregates; however, here, by using our recently developed AS-PLA assay[17,18], we detected α-syn oligomers (Fig. 3e–f). We detected a significant age-dependent accumulation of α-syn oligomers, which likely is linked to the motor defects detected. We therefore decided to use the SNCA-OVX model to test whether CLR01 treatment could affect these phenotypes in vivo.

Mathematical in silico models have been previously used to predict and model pharmacological and biological processes[21–24]. We first administered radiolabeled CLR01 systemically to 12-month-old animals to measure the blood–brain penetration of the compound to inform an in silico model that would allow us to determine whether a monthly dose of 40 µg/kg/day, which was previously used[8], would provide favorable pharmacokinetic profile in the brain. The proportion of CLR01 in the brain was determined as the percentage of radioactivity in the perfused brain, relative to that of the blood, at three different doses and three time points in 12-month-old animals. Our results indicate that 10–35% of the drug concentration available in the blood is present in the brain, and that this is a time- and dose-dependent process (Supplementary Fig. 8A). Based on these data and previously published blood kinetics[14], we first predicted the extra- and intracellular accumulation of the compound in the brain (Supplementary Fig. 8B), then simulated a 2-month treatment spread across 16 subcutaneous injections at 40 µg/kg/day (Supplementary Fig. 8C), to obtain pharmacokinetic parameters (Supplementary Table 1). We found that the intracellular time averaged area under the curve was 240 ng/ml, with a predicted maximum concentration of 510 ng/ml, which resulted in cells being 93% of the experimental time above the $EC_{50}$ (85.4 nM). The model predicted a rapid intracellular accumulation of CLR01, which lead to lower concentrations and time over $EC_{50}$ in the extracellular space, which were still above $EC_{50}$ levels and generally high (extracellular time averaged area under the curve was 100 ng/ml, maximum concentration was 230 ng/ml, and time above the $EC_{50}$ was 81%). This simulation showed that CLR01 would be available at therapeutically relevant concentrations in the extracellular space and the intracellular compartment of the brain.

We then treated transgenic animals with CLR01 using three different paradigms (Supplementary Fig. 9) as follows: first, 4-month-old mice were injected for 2 months subcutaneously, then 12-month-old mice were subcutaneously injected for 2 months, and, finally, 18-month-old animals were treated with a subcutaneous implant of an osmotic mini-pump, to minimize manipulation and stress of aged animals. As an end-point for our 12- and 18-month cohorts, we assessed the animals for both motor behavior and neuropathology, whereas the 6-month-old cohort was evaluated for dopamine release and microglial morphology. The 6-month cohort would be too young to show a neuronal-loss-associated motor phenotype, considering our previous findings[12].

We treated 6-month-old animals to assess whether treatment at an early timepoint was able to change early phenotypes, which

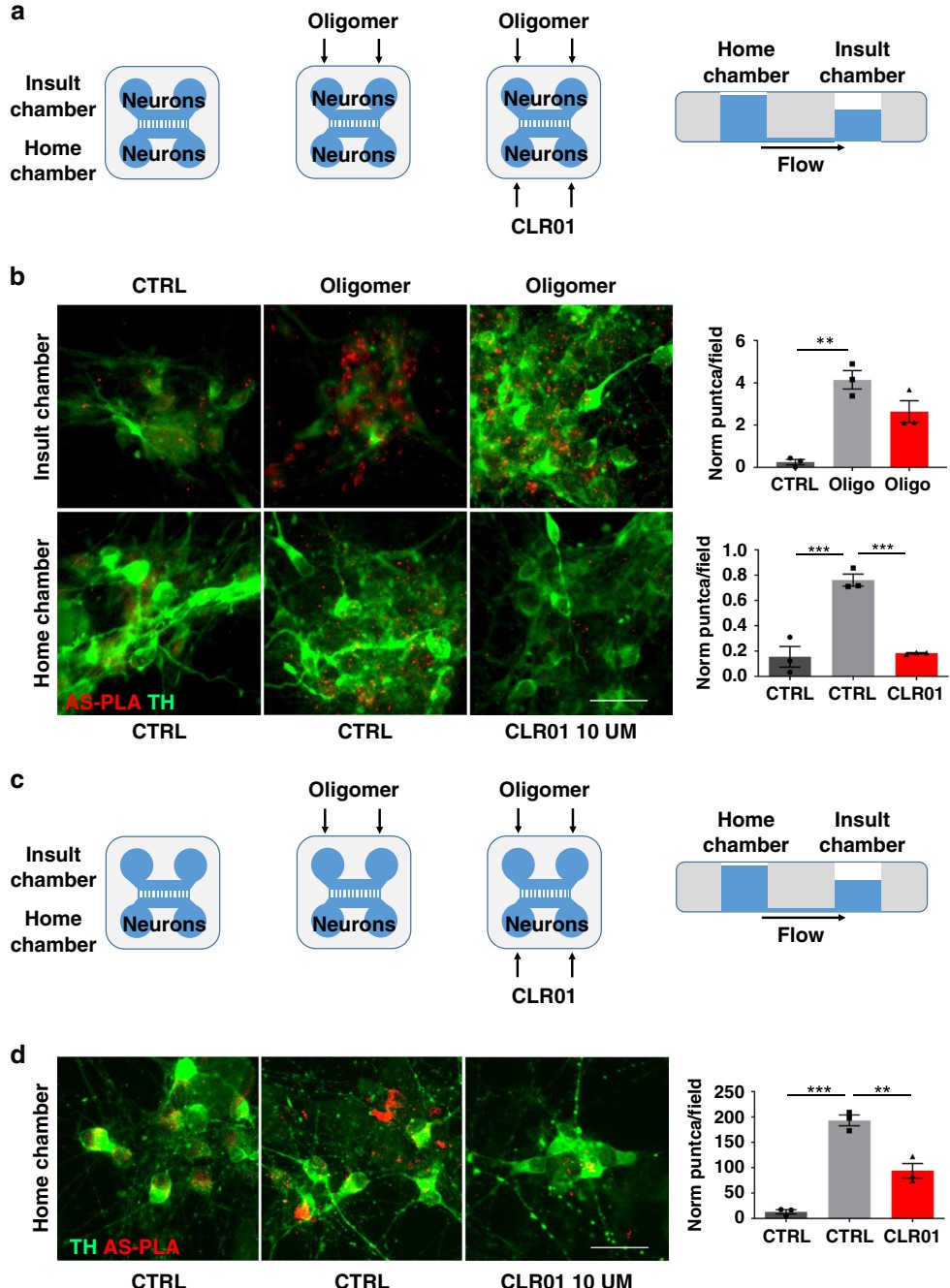

**Fig. 2 CLR01 prevents α-syn aggregation in microfluidic chambers. a, c** Schematic representation of the experimental design for the microfluidic-chamber experiments. **b, d** Representative immunofluorescence images of dopaminergic neurons grown in the microfluidic devices insulted with oligomers and/or treated with CLR01 for 5 days (25 µg/ml and/or 10 µM, respectively). Images were taken at least one field of view away from the microgrooves to avoid artifacts due to contact with the silicone. Scale bars = 25 µm. Quantification of normalized puncta per field of three independent control cell lines analyzed using a one-way ANOVA (Tukey). Oligo: oligomer. **b** $F_{(2, 6)} = 24.31$, $p = 0.0009$ and $F_{(2, 6)} = 39.09$, $p = 0.0005$, $p = 0.0006$. **d** $F_{(2, 6)} = 71.45$. *$p < 0.05$, **$p < 0.01$, ***$p < 0.001$. For all appropriate panels, data are presented as mean values ± SEM.

may be linked to α-syn overexpression rather than α-syn aggregation. At 6 months, dopamine release evoked by discrete electrical stimuli in the striatum in acute slices as measured by fast-scan cyclic voltammetry (FCV) was unchanged (Supplementary Fig. 10A, B). Likewise, we could not detect a difference in α-syn oligomers in dopaminergic cells or microglial morphology (Supplementary Fig. 10C–F), precluding testing treatment effect in these young mice.

These results prompted us to treat older mice to assess whether they would show the expected deficits and allow measurement of

neuroprotection, leading to a rescue of motor function. The 12-month-old *SNCA*-OVX mice showed motor deficits in the latency to fall on a rotarod and speed in catwalk analysis, which was rescued by a twice-weekly subcutaneous injection of CLR01 (Fig. 4a, c), which was accompanied by restorative trends for several other motor phenotypes (Fig. 4b, d and Supplementary Fig. 11). Stool analysis, locomotor activity (LMA), and muscle strength and coordination were unaffected by the treatment (Supplementary Fig. 11). To test whether these effects were specific to *SNCA*-OVX, we dosed a further small cohort of Wt

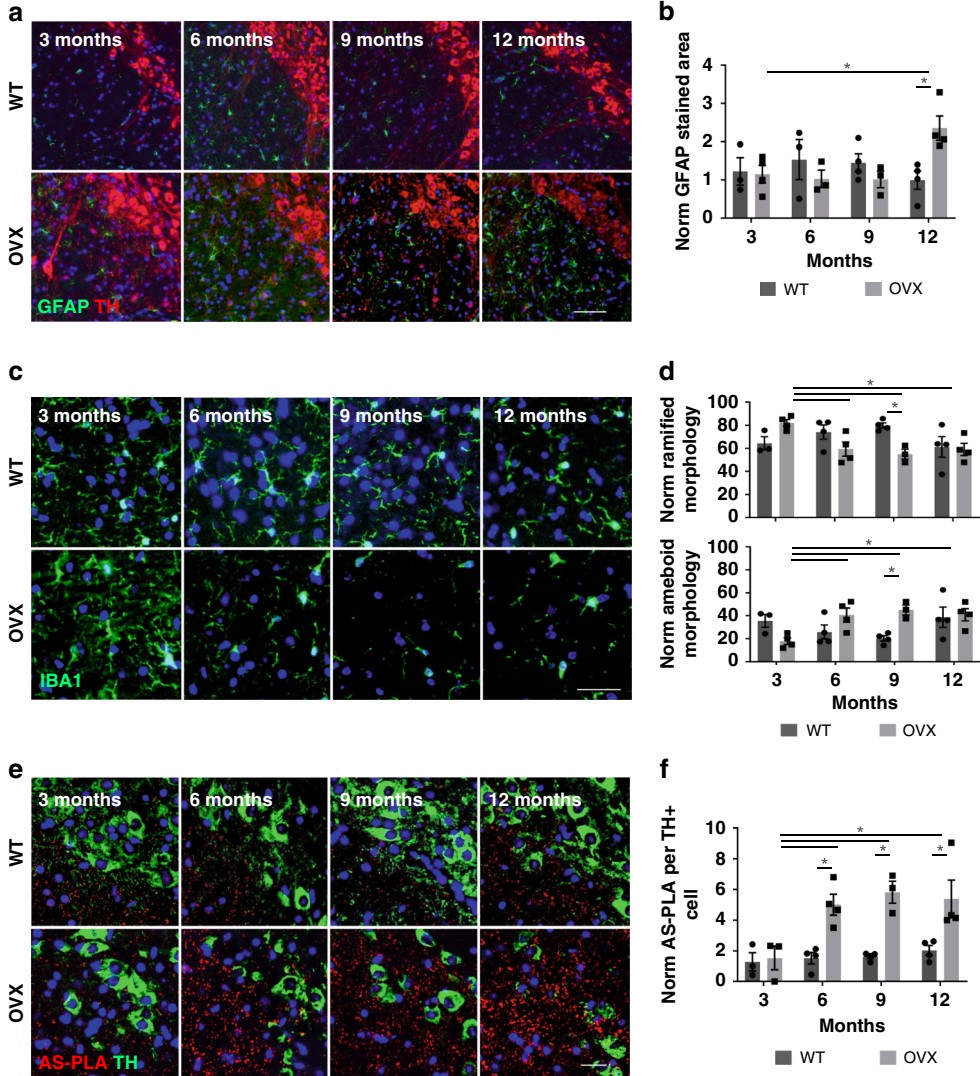

**Fig. 3 Glial activation and α-syn aggregation in *SNCA-OVX* mice. a**, **c**, **e** Representative images of immunofluorescence of GFAP (astrocytes), IBA1 (microglia), and AS-PLA in dopaminergic neurons, respectively. Scale bars = 200, 25, and 25 μm, respectively. **b**, **d**, **f** Quantification of normalized stained area, morphology (expressed as a % from total), and puncta per TH+ cell analyzed using a two-way ANOVA (Dunnett's for age and Sidak's for genotype, for age all values compared to 3 months), $n = 3$–4 animals per group. **b** $F_{(3, 20)} = 1.211$, $p = 0.0177$ for age and $F_{(1, 20)} = 0.1572$, $p = 0.0106$ for genotype. **d** $F_{(1, 22)} = 2.216$, $p = 0.0287$ for genotype and $F_{(3, 22)} = 1.764$, $p = 0.0209$, $p = 0.0104$, $p = 0.0192$ for age. **f** $F_{(3, 21)} = 4.528$ for age, $p = 0.0058$, $p = 0.0016$, $p = 0.0024$ and $F_{(1, 21)} = 33.41$, $p = 0.0043$, $p = 0.0015$, $p = 0.0061$ for genotype. *$p < 0.05$. For all appropriate panels, data are presented as mean values ± SEM. OVX: *SNCA*-OVX.

and *SNCA*-OVX animals with the same dose and paradigm as above and subjected them to rotarod and catwalk analysis. Wt animals showed no changes in their behavior after CLR01 treatment as measured by both rotarod and catwalk (Supplementary Fig. 12), whereas a trend for improvement (which did not reach significance probably due to smaller sample size) was seen in the *SNCA*-OVX animals, suggesting that CLR01 may require the presence of aggregates to elicit its effects. We also observed a significant reduction in α-syn oligomers in dopaminergic SNc cells, and a reduction in GFAP staining (Fig. 4e–h). Interestingly, the morphology of microglia shifted further towards ameboid rather than ramified (Fig. 4i, j), which could be due to a shift towards a phagocytic phenotype to clear the debris generated by CLR01 treatment. Overall, we found that twice-weekly CLR01 injection is able to restore motor behavior and downregulate oligomeric deposition as well as astrogliosis.

Finally, we decided to test the effects of CLR01 treatment in 18-month-old animals. We could not detect any changes in the

behavior or stool analysis of 18-month-old animals carrying a surgically implanted mini-pump releasing CLR01 at a constant rate of 0.11 μl/h (Supplementary Fig. 13). However, we did detect a significant reduction in the number of α-syn oligomers per dopaminergic neuron in the SNc as shown in Fig. 5a, c. This prompted us to investigate whether the treatment had impacted the astrocytic and microglial populations of the SN. We could not detect any changes in the GFAP-positive area or the morphology of Iba1-positive cells in the SN of CLR01-treated animals (Supplementary Fig. 14A–C). These results were unsurprising, as by 18 months *SNCA*-OVX mice have lost ~30% of their SNc dopaminergic neurons and, therefore, a rescue in motor behavior or astrogliosis would be difficult to achieve after neurons are lost. However, CLR01 was able to reduce the α-syn oligomer burden in SN astrocytes and microglia (Fig. 5b, e), and to significantly elevate the levels of PGC1α and Lamp2A in the midbrain of these animals (Fig. 5f, g), but not those of Lamp1 (Supplementary Fig. 14D). PGC1α is a neuroprotective mitochondrial biogenesis

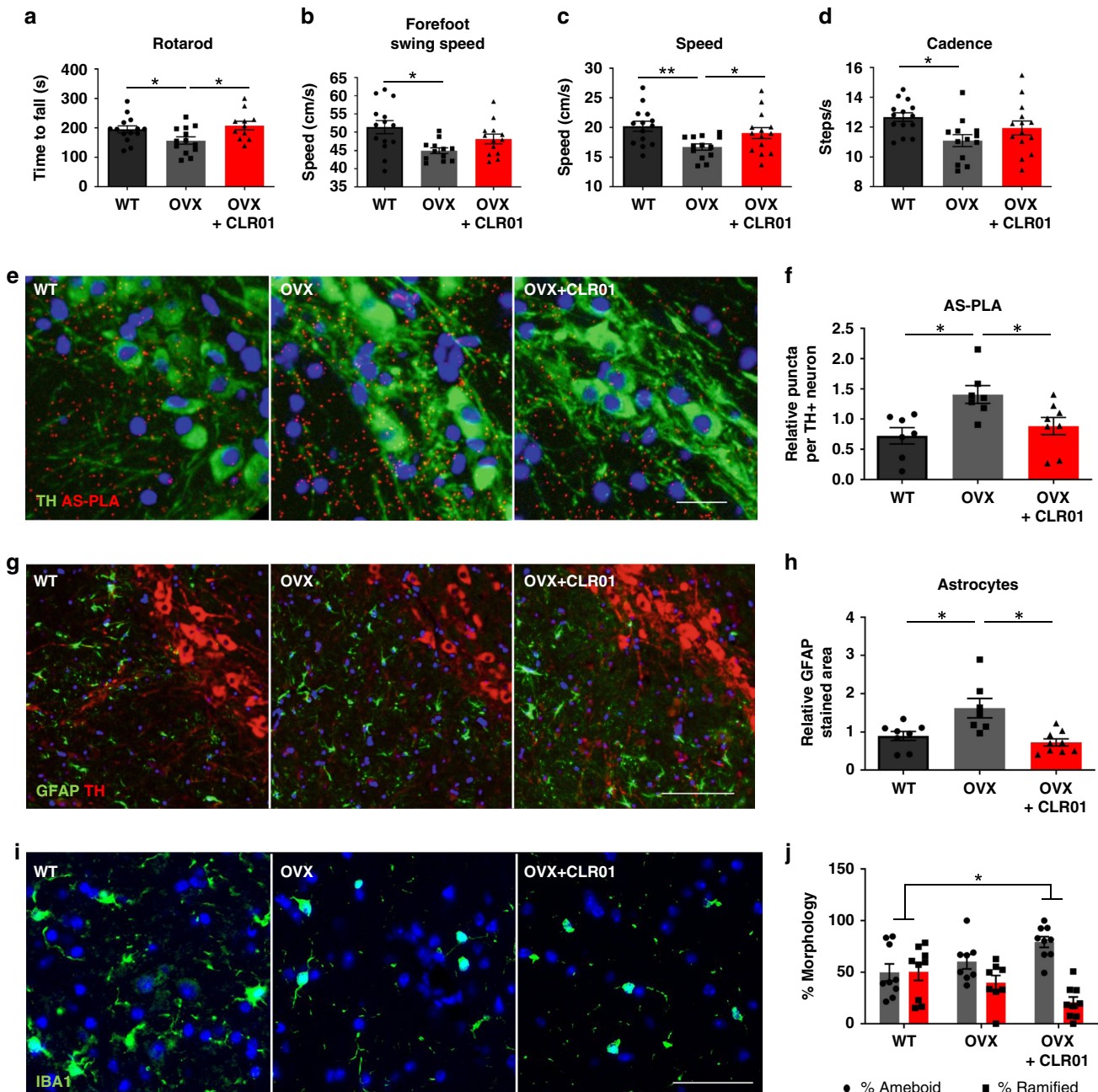

**Fig. 4 CLR01 restores motor behavior and reduces early pathology in vivo at 12 months of age. a–d** Behavior analysis of rotarod (**a**) and catwalk gait analysis (**b–d**) after 2 months of subcutaneous administration of 40 μg/kg/day drug or PBS analyzed using a one-way ANOVA (Sidak), $n = 11$–14 animals per group. **e, f** Representative images and quantification of AS-PLA puncta per TH+ cell. Scale bars = 25 μm. **g–j** Representative images and quantification of GFAP-stained area per field and microglial morphology (expressed as a % from total). Scale bars = 100 and 50 μm, respectively. **a–j** were analyzed using one-way ANOVA (Holm–Sidak), **e–j** $n = 8/8$. CLR: CLR01 *$p < 0.05$, **$p < 0.01$. **a** $F_{(2, 35)} = 4.138$, $p = 0.0376$, $p = 0.0205$. **b** $F_{(2, 37)} = 4.000$, $p = 0.0058$. **c** $F_{(2, 38)} = 4.884$, $p = 0.0078$, $p = 0.0451$. **d** $F_{(2, 38)} = 3.982$, $p = 0.0151$. **f** $F_{(2, 19)} = 6.057$, $p = 0.0072$, $p = 0.0338$. **h** $F_{(2, 21)} = 8.797$, $p = 0.0091$, $p = 0.0012$. **j** $F_{(2, 23)} = 4.750$, $p = 0.0106$. For all appropriate panels, data are presented as mean values ± SEM.

master regulator that plays a role in mitophagy and reactive oxygen species (ROS) reduction, often perturbed in PD[25]; Lamp2a is involved in chaperone-mediated autophagy (CMA) and has been shown to be reduced in post-mortem PD brains[26]. It is interesting to hypothesize that CLR01 might be able to unburden TH+ cells from oligomeric α-syn, thereby indirectly enabling mitochondrial turnover and ROS reduction, and restoring CMA function, which are decreased with aging and disease[25,27,28]. Finally, we confirmed the impact of CLR01 treatment on α-syn aggregates by evaluating midbrain levels of

Triton X-100 and SDS-soluble and -insoluble fractions of α-syn. *SNCA*-OVX showed no signs of SDS-insoluble aggregates (which agrees with the detection of oligomeric pathology), with most of their α-syn content detectable in the Triton X-100 fraction (Fig. 5h, i and Supplementary Fig. 14E). However, a small fraction of α-syn was present in the SDS-soluble fraction in the midbrains of *SNCA*-OVX animals, which was reduced by CLR01 treatment. In conclusion, CLR01 treatment downregulated SN α-syn oligomers and upregulated neuroprotective markers at 18 months of age in *SNCA*-OVX mice.

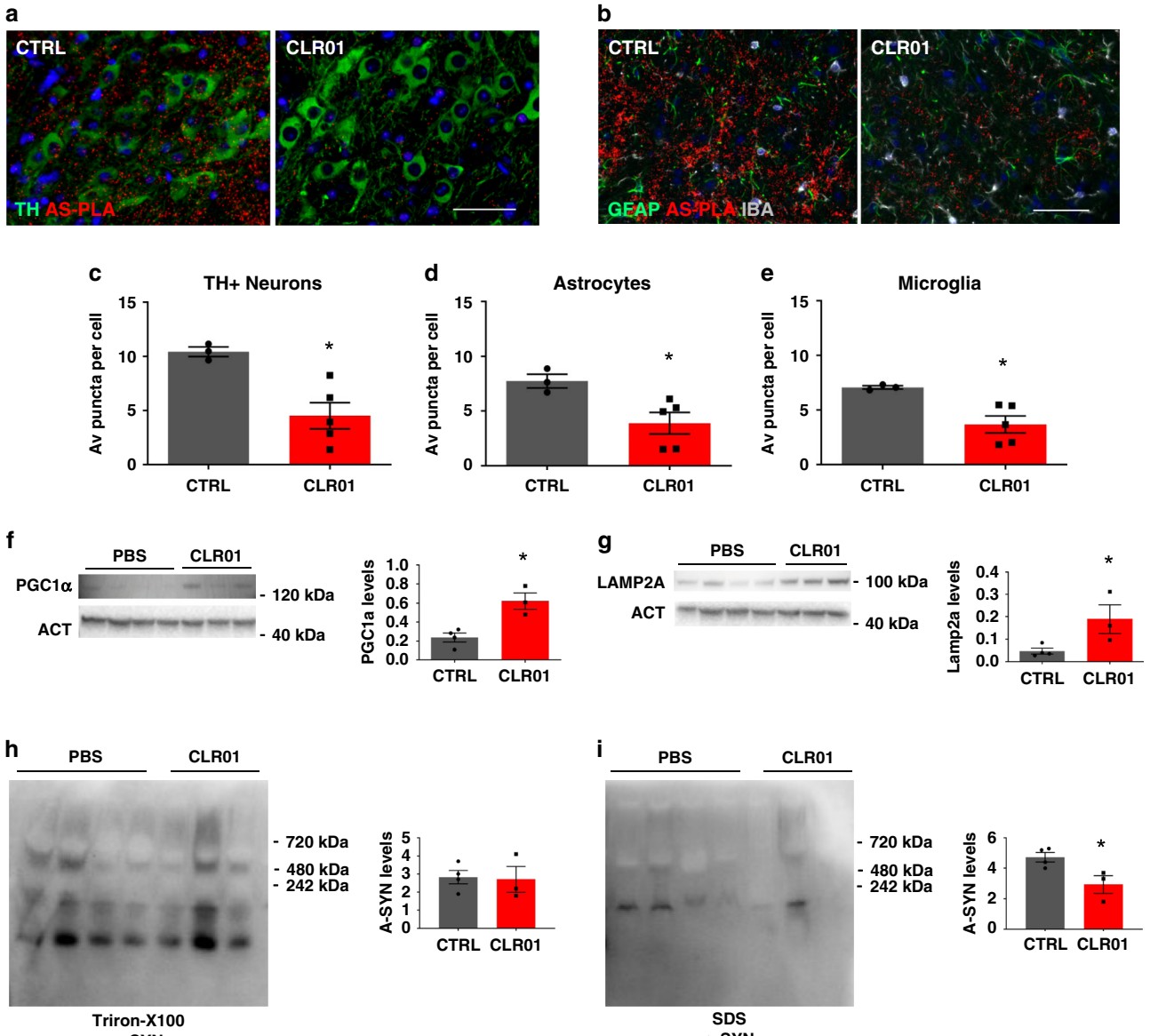

**Fig. 5 CLR01 reduces α-syn aggregation in vivo at 18 months of age. a–e** Representative images and quantification of AS-PLA puncta per TH+, GFAP+, and Iba1+ cells after osmotic mini-pump implantation of 40 μg/kg/days CLR01 or PBS, analyzed using a two-tailed t-test, n = 3/4 independent animals. Scale bars = 50 μm. **c** p = 0.0117, **d** p = 0.0319, **e** p = 0.0179. **f, g** Western blottings of PGC1α and Lamp2a, analyzed using a one-tailed Mann–Whitney U-test, n = 3–4 independent animals. **f** p = 0.0286, **g** p = 0.0286. **h, i** Native western blottings of Triton X-100 and SDS-soluble α-syn, analyzed using a one-tailed Mann–Whitney U-test, n = 3–4 independent animals. **i** p = 0.0286. *p < 0.05. AV average, CLR CLR01. For all appropriate panels, data are presented as mean values ± SEM.

Taken together, these results highlight that CLR01 effects are age-dependent and likely linked to α-syn oligomerization and aggregation, which increase with age, rather than to α-syn overexpression itself.

**CLR01 reduces α-syn aggregation after intracranial aggregate injection**. We tested whether CLR01 could reduce protein pathology using two different intracranial injection models (Supplementary Fig. 9), representing more late-stage pathology than that of *SNCA*-OVX mice, which are a model for early oligomeric accumulation of α-syn. First, we used a recombinant mouse α-syn pre-formed fibril (mPFF) injection model, which was previously described by Luk et al.[29]. We injected 3-month-old C57Bl/6 WT mice with mPFF preparations in the dorsal striatum and then followed pathology in the SNc of these mice.

Upon inspection of the SNc, lesions labeled by phospho-α-syn were found to be more prevalent in the SNc than in the ventral tegmental area, as was expected following dorsal striatum injection (Supplementary Fig. 15A–D). Subcutaneous injections of CLR01 twice-weekly reduced the number of lesions in the SNc compared to animals treated for a month with phosphate-buffered saline (PBS) (Supplementary Fig. 15A, B).

We next tested whether CLR01 could reduce pathology induced by the injection of LB extracts as described by Recasens et al.[13]. LB extracts were injected into the SN of 4-month-old C57Bl/6 WT mice, which were aged for a further 3 months before CLR01 treatment. As these extracts are toxic and can lead to neurodegeneration, we decided to implant an osmotic mini-pump to deliver either PBS or CLR01 for the last month to reduce manipulation and stress for the mice. Stereology and

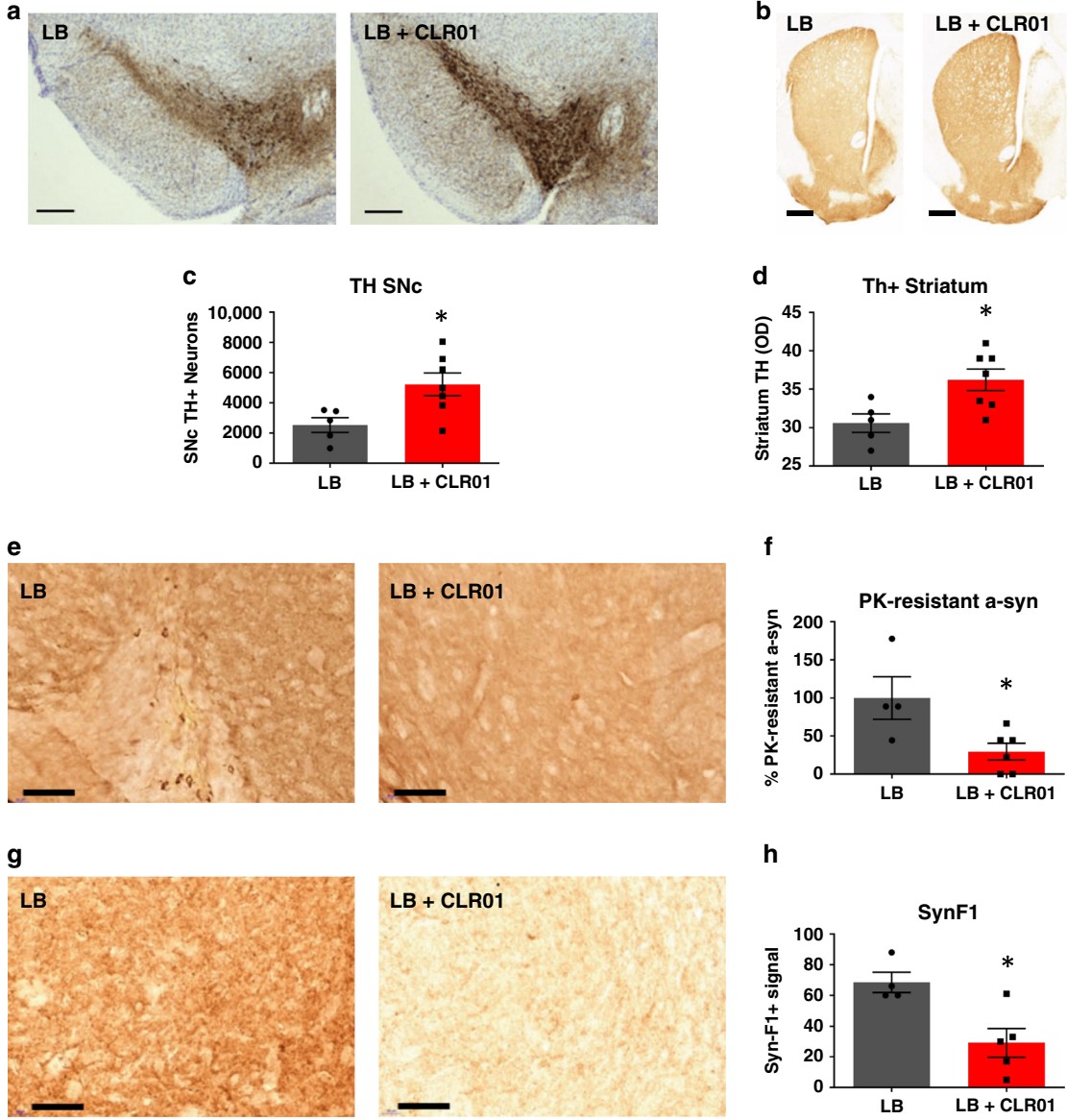

**Fig. 6 CLR01 protects dopaminergic neurons from LB-induced cell death through reduction of α-syn aggregation in mice. a, b** Representative photomicrographs of TH-immunostained SNc and striatum (respectively) in LB-inoculated mice after osmotic mini-pump implantation for CLR01 (or PBS) delivery (40 μg/kg/h). **c** Stereological cell counts of SNc TH-immunoreactive neurons (**a**) in LB-inoculated mice, at 3 months post-LB inoculation. $p = 0.0177$. **d** Optical densitometry of striatal TH immunoreactivity in LB-inoculated mice (**b**), at 3 months post-LB inoculation. $p = 0.0354$. **e, g** Representative photomicrographs of α-syn and PK-resistant α-syn immunostained SN from LB-inoculated mice, and corresponding quantification. **f, h** Representative images of Syn-F1 immunostaining in the SN in LB-inoculated mice and corresponding quantification of the intensity levels of Syn-F1 staining. **f** $p = 0.0333$, **h** $p = 0.0556$. **a–h** $n = 4–5$ for LB-inoculated mice, $n = 5–7$ for LB + CLR01-independently treated animals, analyzed with a two-tailed Mann–Whitney U-test. *$p < 0.05$. Scale bars = 0.1 mm (SN) and 1 mm (Str). For all appropriate panels, data are presented as mean values ± SEM.

densitometry analysis showed that CLR01 treatment was able to rescue TH+ immunoreactivity in the SN and striatum, respectively (Fig. 6a–d). Nissl+ cell counts in the SN and DAT in the striatum were only partially restored (Supplementary Fig. 15E, F). Further analysis of the midbrains of these mice showed that there was a reduction in α-syn PK-resistant puncta and a reduction of staining by the Syn-F1 antibody, which are indicators of a reduction of aggregated α-syn (Fig. 6e, h).

## Discussion

Neuroprotective therapies targeting α-syn in PD have a high potential for translation to patients in the clinic. Working across a range of human in vitro and rodent in vivo α-syn-based models of PD, we have shown that CLR01 reduces α-syn aggregation and

alleviates disease-related phenotypes. CLR01 dismantles synthetic oligomeric and fibrillar, and patient LB-derived α-syn, alleviating their toxic effects in vitro, reducing neurodegeneration-associated α-syn accumulation in vivo, and correcting behavioral phenotypes in a transgenic mouse model of PD.

Applying patient LB extracts treated with CLR01 to the highly physiological human iPSC-derived dopaminergic cultures in vitro gives a fully humanized model of disease. The LB extracts show a higher α-syn oligomeric content than noLB extracts. CLR01 reduced the α-syn oligomeric content and with it the toxic effect of the patient LB extracts. However, LB extracts likely contain many other proteins[30–32], which could be sensitive or resistant to CLR01 treatment. When the extracts were applied to iPSC-derived dopaminergic cultures, we observed that CLR01 potently

inhibits the toxic effects of LB extracts in both iPSC-derived and primary neuron culture systems, as previously shown in the case of Aβ-induced LTP defects in primary mouse hippocampal and cortical neurons, and in an AD mouse model[8].

Microfluidic chambers facilitate the isolation of axonal terminals from cell bodies in culture modelling pathology along the neuron from synapse to soma. Several studies have shown that degeneration may start in the striatal axons before affecting the cell soma of dopaminergic neurons[33]. In support of this theory, neuroimaging studies have shown that human PD presents early with a larger loss of transporters and other DA axon markers in the striatum than in the SNc[34,35]. As a result several groups have studied this early degeneration in α-syn overexpression mouse models and shown early striatal synaptic and dopamine release defects prior to neurodegeneration[12,36]. It is therefore important to investigate in vitro and in vivo whether the toxic effects of α-syn on axon terminals can be alleviated with CLR01 administration. Using microfluidic chambers in vitro, we observed that CLR01 reduced the accumulation of α-syn oligomers in the cell soma after treatment of axonal terminals, although it is possible that this was in part due to an inhibition of retrograde transport of α-syn, or to the minimal flow of CLR01-containing medium to the insult chamber. In parallel studies in vivo, CLR01 administration reduced lesions in dopaminergic cell somas in the SNc after injection of mPFFs to the dorsal striatum, confirming a protective effect of CLR01 against α-syn pathology in both axons and cell bodies.

The age-dependent microglial and astrocytic activation in the *SNCA*-OVX mouse model overexpressing α-syn supports a role for inflammation in PD and other neurodegenerative disorders[37–39]. The observation that microglial activation occurs in a time-dependent manner almost simultaneously to oligomer deposition, and that astrocyte activation is delayed in comparison, is consistent with studies showing that astrocytic activation is induced by microglial activation and is mediated at least in part by Il1α, TNF, and C1q[40]. In *SNCA*-OVX mice at 6 months, the overexpression of α-syn causes only mild oligomer accumulation and disease-related motor phenotypes were not observed. The window for treatment with a molecular tweezer is optimal at twelve months, when early disease processes are underway and neuronal dysfunction has begun, but before extensive and widespread dopaminergic neuron degeneration. By 18 months, when dopaminergic neurons have already been lost and neuroinflammation has been manifest for the previous 6–12 months, inhibition of α-syn self-assembly by CLR01 is unable to reverse cellular and behavioral phenotypes. This highlights the importance of studying the disease in a late-onset age-dependent model and of treating the disease before it has progressed beyond a putative point-of-no-return. Our work also demonstrates that continuous administration, as used previously in the rapidly-progressing Thy-1 α-syn overexpression model of PD[41], is unnecessary, further supporting the clinical development of CLR01 in PD patients. We have also found that CLR01 is able to elicit its function in the nanomolar range, which makes it a potent compound that can be effective even at low doses, such as the ones we use in vivo. These low doses showed promising predicted pharmacokinetic properties in silico, which support previous findings[8,14] and our in vivo results.

Finally, to extend the application of CLR01 beyond transgenic strains and to models of protein aggregate burden, we demonstrated a positive therapeutic effect of CLR01 against α-syn-induced pathology after intracranial injection of LB extract into the SN. CLR01 treatment exhibited a neuroprotective effect demonstrated by reduced α-syn pathology and a rescue of dopaminergic neurons in the SNc and increased striatal TH+ innervation in the striatum. The positive effect of CLR01 in vivo in both late-onset, age-dependent models, and in rapid models of accelerated protein pathology demonstrates the robust application of the molecule across diverse disease models.

We have investigated the broad utility of CLR01 against different forms of aggregated α-syn which complements previous work on other amyloid assembly inhibitors. For example, epigallocatechin-3-gallate (EGCG) has been shown to bind α-syn inhibiting its aggregation[42,43]. It has also been shown to be protective in vitro[44] and has shown promise by restoring MPTP-induced neuronal loss[45] and reducing peripheral immune response after MPTP treatment[46] in mouse models. The relatively small size of CLR01 allows it to be internalized into cells as has been shown in our study and by others[47], providing an advantage over antibody based therapies, which generally target external epitopes, and are therefore unable to fully tackle toxic intracellular α-syn species. Taken together, these data in diverse in vitro and in vivo preclinical models suggest that molecular tweezers are highly promising candidates for the treatment of PD.

## Methods

**Study design**. Sample size was determined according to the experimental paradigm. Sixteen animals were bread to account for loss of animals during ageing and avoid $n < 10$ at the end of the study where behavioral differences of ageing cohorts were investigated. Any animals showing any welfare issues were taken off the study, otherwise animals remained in the sample. For neuropathological examination four to five animals were bred per condition in order to avoid $n < 3$. For iPSC culture, three to four independent control lines were used in order to ensure $n \geq 3$. For biochemical analysis of recombinant or biological material, three independent experiments were performed on three independent samples.

Treatments were administered blindly by coding of the different groups into numbers. Samples were blinded by taping over identifiers, to avoid bias when sampling. For EM imaging, four independent fields were imaged and a representative image was shown. For cell imaging, three to nine independent fields of view were analyzed per well and one to two wells were analyzed per condition and control line. For tissue analysis, two to four independent fields (i.e., at least 100 μm apart to avoid re-sampling) of anterior SN were analyzed, and for puncta counts associated with TH+ cells, 25 neurons were sampled across the SNc. For blood–brain barrier experiments, one radioactivity value was obtained per brain and animal. For behavior and stereology, sampling is described in their particular methods section.

**Electron microscopy**. Ten microliters of 0.2 mg/ml sample were applied to glow-discharged, carbon-coated EM grids (TAAB). Samples were incubated for 2 min on the grids at room temperature (RT), followed by 2% uranyl acetate staining for 10 s. Samples were rinsed with water, dried, and stored at RT for analysis. Images were acquired with a FEI Tecnai 12 transmission EM (120 kV) with a Gatan US1000 camera.

**α-Syn aggregation**. α-Syn (2 mg/ml; Peptide) were shaken at 37 °C and 250 r.p.m. for as many days as described in each case, to induce aggregation of α-syn. To produce oligomers, α-syn was shaken at 1 mg/ml and for 3 days (Supplementary Fig. 16A).

**PLA, immunofluorescence, and immunohistochemistry**. AS-PLA experiments were performed using the Duolink kits (Sigma) according to the manufacturer's instructions[17,18]. An α-syn antibody (mouse monoclonal anti-α-syn4D6, ab1903, Abcam, 1:2000), was used to prepare conjugates using Duolink Probemaker Plus and Minus kits. For α-syn-dynein and α-syn-kinesin interaction experiments, we used the same α-syn antibody with monoclonal antibodies targeting dynein and kinesin (mouse monoclonal anti-dynein MAB1618 and anti-kinesin MAB1614 from Millipore, respectively, both 1:100). For fluorescent AS-PLA of brain extracts and cell samples, samples were fixed in 4% paraformaldehyde (PFA). All samples were incubated with Duolink block solution at 37 °C for 1 h and then with α-syn conjugates diluted in Duolink PLA diluent overnight (ON) at 4 °C. Samples were washed with tris buffered saline (TBS) containing 0.05% Tween-20 (TBS-T) and incubated with Duolink ligation reagents for 1 h at 37 °C, washed four times with TBS-T, and then incubated with Duolink amplification reagents for 2.5 h at 37 °C. Samples were washed and then mounted in FluorSave (Calbiochem).

Paraffin-embedded tissue was dewaxed by 2 min consecutive incubations in Xylene, Histoclear, 100% ethanol, 95% ethanol, 70% ethanol, and H$_2$O. After rehydration, samples were incubated in 10% H$_2$O$_2$ in PBS, to reduce background and heated in a microwave in citrate buffer pH 6.0 (Abcam) for antigen retrieval. After antigen retrieval, samples were subjected to immunofluorescence or AS-PLA as necessary.

For AS-PLA co-immunofluorescence, immunofluorescence was performed after antigen retrieval and before PLA block. This process consisted of 1 h RT incubation in 10% Serum TBS-T block, 1 h RT incubation in primary antibody (GFAP Z0334 Sigma 1:1000, Iba1 019-19741 Wako 1:1000, TH ab152 Millipore 1:1000, TH ab76442 Abcam, 1:100, Tuj1 ab107216 Abcam, 1:100), after which slides were washed with TBS-T. Slides then were incubated for 1 h at RT with secondary antibodies (Alexa488, Alexa594, or Alexa680, Life Technologies), after which they were washed again with TBS-T. If AS-PLA analysis was required, samples went on to AS-PLA block, if not samples were mounted with FluorSave.

For recombinant protein analysis, 10 μl of 2 mg/ml samples were spotted on poly-L-lysine coated cover slips, left at RT for 30 min, then PFA-fixed for 10 min, and finally treated as above for fluorescent AS-PLA analysis.

**Histopathological analysis of cryoprotected tissue**. Extent of lesion: To assess the integrity of the nigrostriatal pathway, TH immunohistochemistry was performed on 50 μm free-floating SNc and striatal sections. Briefly, sections from three representative levels of the striatum (anterior, medial and posterior) and serial sections (1/6) corresponding to the whole SNc were incubated with a rabbit monoclonal antibody raised against TH (Abcam, EP1532Y, 1:5000) for one night at RT and developed with an anti-rabbit peroxidase EnVisionTM system (DAKO, K400311) followed by 3,3′-diaminobenzidine (DAB) visualization. Free-floating SNc sections were mounted on gelatinized slides, counterstained with 0.1% cresyl violet solution, dehydrated, and cover-slipped, whereas striatal sections were mounted on gelatinized slides and cover-slipped. The extent of the lesion in the striatum was quantified by optical density (OD). Sections were scanned in an Epson expression 10000XL high-resolution scanner and images were used in ImageJ open source software to compare the grey level in the putamen. TH-positive SNc cells were counted by stereology, blind with regard to the experimental condition, using a Leica DM6000B motorized microscope coupled with the Mercator software (ExploraNova).

α-Syn pathology: Synucleinopathy was assessed using a primary antibody that recognizes both mouse and human α-syn (mouse anti-α-syn monoclonal antibody, BD Transduction Laboratories, clone 42, #610787; BD Biosciences, SYN1, 1:1000) and aggregated α-syn antibody (mouse anti-α-syn monoclonal antibody, BioLegend, Syn-F1, 1:2000). In addition, SN sections were first incubated with or without proteinase K (PK) at 1 μg/mL (Sigma-Aldrich) in PBS for 10 min at room temperature[13,48]. PK-treated and nontreated adjacent sections then were incubated ON at room temperature with SYN1 antibody, 1:1000. The following day, they were incubated with a secondary antibody conjugated to a peroxidase EnVision system (DAKO) DAB incubation. Sections then were mounted on gelatinized slides, dehydrated, counterstained if necessary and cover-slipped until further analysis. Images were used in ImageJ to count PK-resistant α-syn-positive dots in SN (SYN1) or to measure OD in SN (Syn-F1). For striatal injections, tissue was paraffin embedded and processed as specified above. The whole SN was sectioned at 5 μm and every tenth section was stained for p-α-syn (EP1536Y, Abcam) and TH (sc-25269, Santa Cruz). The sections with most p-α-syn+/TH+lesions were blindly selected, and the resulting number of lesions was averaged for independent animals.

**Imaging**. Samples were imaged with the Opera Phenix confocal microscope and Harmony software (Perkin-Elmer) or the EVOSflAUTO (Thermo). For standard cell culture, images were automatically captured with the Opera Phenix and analyzed with ImageJ or Cell Profiler for the detection of puncta (AS-PLA signal) or cell nucleus/body (TO-PRO3, DAPI, TH counts), respectively. To analyze process morphology, average length and number of puncta were analyzed. For microfluidic and tissue samples, images were taken blinded to treatment on the EVOSflAUTO. We used automatic measurement with ImageJ for puncta counts (AS-PLA signal) or stained areas (GFAP signal). To count PLA puncta inside cells of interest, puncta were counted blindly by eye in 25 cells per animal. In order to determine microglial morphology, all microglial cells in the SN were clasified by a blinded operator by eye as ramified or ameboid.

**iPSC-derived dopaminergic cultures**. iPSCs were cultured according to the methods described in Supplementary Information and previous reports[9,11,49].

NoLB and LB brain extracts were purified as described below and treated for 10 days ±10 μM CLR01 (Supplementary Fig. 16B) with continuous shaking at 37 °C and 250 rpm. Cells were treated with 600 pg/ml noLB/LB extract ±10 μM CLR01 as indicated on DIV36 and DIV38, and then harvested and measured on DIV41. To culture cells in microfluidic chambers, dual axonal separation microfluidic devices were purchased from Millipore and adapted for dopaminergic cell survival. Briefly, the cell channel was carefully opened with a scalpel without disturbing the microgrooves for axonal separation. The device was tested ON by leaving the chambers filled with unequal volumes; if the volumes remained unequal after ON incubation the chambers were used for experiments. For microfluidics assays, CLR01 was used at 10 μM and oligomers were applied at 25 μg/ml.

For live-dead staining, TO-PRO-3 was added to culture medium 1:1000 (Thermo).

**Neuroblastoma cell cultures**. Neuroblastoma SH-SY5Y cell cultures were cultured in Dulbecco's modified Eagle's medium supplemented with 10% fetal bovine serum (FBS), 1% penicillin/streptomycin, and 1% L-glutamine (Life Technologies) at 37 °C and 5% $CO_2$. On DIV0 cells were plated at 10,000 cells/well on half-area 96-well plates (Greiner). On DIV1 cells were transfected with a pcDNA3.1 vector containing human α-syn (Addgene) with Lipofectamine 2000 and Plus reagent (Life Technologies) in OptiMem (Life Technologies). On the next day, the medium was changed to CLR03-/CLR01-containing medium. Plates were fixed on DIV4 and then processed for AS-PLA analysis.

**Purification of LBs from human PD brains**. The human brain samples were obtained from brains collected in a Brain Donation Program of the Brain Bank "GIE NeuroCEB" run by a consortium of Patients Associations: ARSEP (Association for Research on Multiple Sclerosis), CSC (cerebellar ataxias), France Alzheimer, and France Parkinson. The consents were signed by the patients themselves or their next of kin in their name, in accordance with the French Bioethical Laws. The Brain Bank GIE NeuroCEB (Bioresource Research Impact Factor number BB-0033-00011) has been declared at the Ministry of Higher Education and Research and has received approval to distribute samples (agreement AC-2013-1887). Human SNc was dissected from fresh frozen post-mortem striatal samples. Tissue was homogenized in 9 vol (wt/vol) of an ice-cold MSE buffer (10 mM MOPS/KOH pH 7.4, 1 M sucrose, 1 mM ethylene glycol tetraacetic acid, and 1 mM ethylenediaminetetraacetic acid) with protease inhibitor cocktail (Complete Mini) with 12 strokes of a motor-driven glass/Teflon homogenizer. For LB purification, a sucrose step gradient was prepared by overlaying 2.2 M with 1.4 M and finally with 1.2 M sucrose in volume ratios of 3.5:8:8 (vol/vol)[50]. The homogenate was layered onto the gradient and centrifuged at 160,000 × g for 3 h using a SW32.1 rotor (Beckman Coulter). Twenty-six fractions (500 μL each) were collected from each gradient from top (fraction 1) to bottom (fraction 26) and were analyzed for the presence of α-syn aggregates by a filter retardation assay as previously described[13,51]. LB-containing fractions from PD patients were those between fractions 21 and 23. NoLB-containing fractions (i.e., fraction 3, at the beginning of the 1.2 M interface) derived from the same PD patients (which contain soluble or finely granular α-syn) but lack large LB-linked α-syn aggregates were obtained from the same sucrose gradient purification. The amount of α-syn in the LB fractions was quantified using a human α-syn ELISA kit (#KHB0061). Further characterization of LB fractions was performed by immunofluorescence analysis. Briefly, LB fractions from the sucrose gradient were spread over slides coated with poly-D lysine and fixed with 4% PFA in PBS for 30 min. Fixed slides were stained with 0.05% thioflavin S for 8 min and then washed three times with 80% EtOH for 5 min, followed by two washes in PBS for 5 min. Finally, all samples were washed 3 times with PBS and blocked with 2% casein and 2% normal goat serum for 30 min. For immunofluorescence analyses, samples were incubated with a human α-syn-specific antibody (clone syn211, Thermo Scientific, 1:1000) for 30 min, washed three times with PBS, incubated with a goat anti-mouse TRITC-conjugated antibody (Jackson, 1:500), before being cover-slipped for microscopic visualization using fluorescence mounting medium.

**Primary rodent cultures**. Cortical neurons were obtained from the cortical lobes of E18 Sprague–Dawley rat embryos: selected cortical tissue was digested with 0.25% trypsin and 0.004% deoxyribonuclease in Hank's balanced salt solution (HBSS; Sigma-Aldrich) for 5 min at 37 °C. The reaction was stopped by adding Neurobasal medium (Invitrogen) supplemented with 10% FBS, B27 (Invitrogen), 2 mM glutamine, and antibiotic-antimycotic mixture, centrifuged at 1000 r.p.m. for 5 min, and the cell pellet was resuspended in 1 ml of the same solution. Mechanical dissociation was performed by using 23-, 25-, and 27G-gauge needles, and the resulting cell suspension was filtered through a 40 μm nylon mesh (Millipore)[52]. Neurons were seeded onto poly-L-ornithine-coated glass cover slips (12 mm) at 10,000 cells/cm². After 24 h, the medium was replaced with serum-free, B27-supplemented Neurobasal medium and maintained at 37 °C and 5% $CO_2$. The cultures were essentially free of macroglia and microglia.

Primary cultures of cerebral cortical astrocytes were prepared from newborn (P0–P2) Sprague–Dawley rats: cortical lobes were extracted and enzymatically digested with 400 μl of 2.5% trypsin and 40 μl of 0.5% deoxyribonuclease in HBSS for 15 min at 37 °C. The reaction was stopped by adding Iscove's Modified Dulbecco's Medium (IMDM) supplemented with 10% FBS (Gibco) and centrifuged at 1200 r.p.m. for 6 min. The cell pellet was resuspended in 1 ml of the same solution and mechanical dissociation was performed by using 21- and 23G-gauge needles[53]. After 2 weeks, cells were trypsinized and astrocytes were plated (15,000 cells/cm²) onto poly-lysine-coated glass cover slips (12 mm).

**Apoptosis assays**. Rat astrocytes and neurons were plated respectively on poly-lysine and poly-ornithine, and then treated with LB (120 pg/ml), CLR01 (10 μM), or with the combination of LB and CLR01. Where indicated, LB was preincubated for 10 days with 10 μM CLR01 in agitation before treating the cells. After 5 days of incubation of the cells with the treatment, cells were fixed with 4% PFA and stained with DAPI (Molecular probes). Images were taken with the Zeiss AxioVision microscope. Apoptosis was calculated as a percentage of nuclei with condensed chromatin vs. total nuclei.

**Transgenic animal cohorts**. Transgenic *SNCA*-OVX mice were bred in house[12], while control animals C57/Bl6 were obtained from Charles River. All animals were house in open top cages (five animals per cage maximum) at 21 °C, 55% humidity in a 06:00 half-light, 07:00 on–18:00 half-light, and 19:00 off dark/light cycle. All procedures were conducted in accordance with the United Kingdom Animals (Scientific Procedures) Act of 1986 and approved by the local ethical review panel at the Department of Physiology, Anatomy and Genetics, University of Oxford, or in accordance with in accordance with the European Union directive of 22 September 2010 (2010/63/EU) on the protection of animals used for scientific purposes, with approval from the Institutional Animal Care and Ethical Committee of Bordeaux University (CE50, France) under the license number 5012099-A. Animals were age, sex, and weight matched.

Osmotic mini-pumps (model 1004 Alzet) were subcutaneously implanted on the back of the neck of each 17-month-old animal. Briefly, animals were anesthetized with 1–2% isoflurane and the pump was implanted through a minimal incision, which was subsequently sutured. The pumps contained CLR01 in sterile PBS (40 µg/kg per day) or PBS as a vehicle. After a month, mice went through behavior testing.

Twelve- and 4-month-old animals were subcutaneously injected every Monday and Thursday for 2 months. To achieve 40 µg/kg per day, the total amount required for a given mouse for the 2 months was divided into 16 × 100 µl doses. After the treatment was complete the mice went through behavior testing or FCV.

Tissues were harvested by cervical dislocation, dissection and flash freezing with dry ice, or perfusion fixing with 4% PFA (Sigma). Fixed tissue was left in PFA ON at 4 °C, and then transferred to 70% ethanol for an additional ON at 4 °C. The tissue was dehydrated and included in wax through 1 h RT incubations as follows: 95% ethanol, 100% ethanol, 100% ethanol, Histoclear (National Diagnostics), Histoclear, wax (Paraplast Plus, Sigma), wax and wax. Sections were then cut using a microtome at 5 µm.

For blood–brain barrier penetration, experiments were performed according to the procedures described in Attar et al.[14]. As subtraction of blood-born radioactivity was proven to be as effective as perfusion, we used this method to correct for blood-born radioactivity in the brain.

**Animals and stereotactic Inoculations**. Wild-type C57Bl/6 mice (4 months old) received bilaterally 2 µl of human PD-derived α-syn (LB fractions, prepared as described above[13]) by stereotactic delivery to the region immediately above the SN (−2.9 mm anteroposterior, ±1.3 mm lateral, and −4.5 mm dorsoventral) at a flow rate of 0.4 µl/min. Three months later, CLR01 was administered subcutaneously to mice using osmotic mini-pumps for 28 days (40 µg/kg per day, Alzet Model 1004). Filling, priming, and implantation of the pumps were performed using standard protocols provided by the manufacturer. After 28 days of treatment, mice were perfused transcardially with a 0.9% saline solution. Brains were removed quickly after death and post-fixed for 3 days in 4% paraformaldehyde at 4 °C, cryoprotected in PBS containing 30% sucrose, and stored at −80 °C until sectioning.

For striatal injections, WT, 3-month-old, C57Bl/6 mice received a 1.5 µl injection of mPFF that had been previously sonicated for 1 min at bregma 0.5 mm anteroposterior, −2.5 mm mediolateral, and 2.7 mm dorsoventral at a flow rate of 200 nl/min with a 32 gauge Hamilton syringe (Hamilton) that was withdrawn 5 min post injection. The mice were subsequently injected with CLR01 subcutaneously every Monday and Thursday for 1 month. In order to achieve 400 µg/kg per day, the total amount required for a given mouse for the month was divided into 8 × 100 µl doses.

**Behavior analysis and FCV**. Spontaneous LMA was assessed by placing each mouse in a transparent plastic cage, and assessing locomotion by means of infra-red beam breaks over a 4 h period in 30 min intervals (apparatus manufactured by San Diego Instruments). The floors of the cages were covered by a thin layer bedding and measurements were started at 9:00.

To analyze gait, we used a CatWalk automated gait analysis system (Noldus Information Technology). Mice were placed on a transparent glass platform illuminated with green light on a red background. Runs were acquired and accepted for analysis if they were between 0.5 and 5 s in duration and there was <35% variation in speed. The first five compliant runs were analyzed with Catwalk software, and the average data for each mouse was taken for analysis.

For rotarod analysis, mice were gently placed on an open rotarod (Med Associates, Vermont, USA) facing away from the experimenter. Mice were tested under the same accelerating conditions (4–40 r.p.m. over 5 min) for two consecutive days starting at 9:00. The first day was considered acclimatization; on the second day, the latency to fall was recorded for analysis.

For muscle strength and coordination analysis, mice were placed on a 43 cm$^2$ wire mesh attached to a 4 cm deep wooden frame. After 5 s, the grid was carefully inverted and the latency to fall was measured. After 60 s, the test was stopped, as this was considered maximum score.

For stool collection animals were placed into separate clean cages with no access to food or water, and fecal pellets were collected over a 1 h period (1600–1700 h). Pellets were weighed to obtain wet stool weight, dried ON at 65 °C, and reweighed to obtain dry stool weight and stool water content.

For FCV animals were culled by cervical dislocation and 300 µm coronal slices were prepared for analysis in ice-cold artificial cerebrospinal fluid (aCSF). Samples were maintained in aCSF saturated with 95% $O_2$/5% $CO_2$ and analyzed through

probing of the dorsal or ventral striatum with a carbon fibre microelectrode. Electrical stimuli were single pulses, repeated at intervals of 2.5 min, delivered at 0.6 mA, for 200 µsec[12]. Data were acquired and analyzed with Axoscope 11.0 (Molecular Devices).

**Western blotting**. Dissected brain regions were lysed in RIPA buffer (SDS-poly-acrylamide gel electrophoresis (PAGE)) or non-denaturing lysis buffer (native PAGE). For immunoblot analysis, 10 µg of total protein per lane was loaded on 4–15% gels (Bio-Rad) and blotted onto polyvinylidene difluoride membranes (Bio-Rad). For SDS-insoluble 0.5 µg of total protein was loaded. Blotted samples were probed with antibodies against α-syn (MJFR1 Abcam), PGC1α (Santa Cruz Biotechnologies sc-13067), Lamp2A (Abcam, ab18528) and Lamp1 (Abcam, ab24170), all 1:1000. After ON incubation with primary antibody at 4 °C blots were washed in TBS-T and incubated in secondary horseradish peroxidase antibody (Bio-Rad, 1:5000) for 1–2 h at room temperature. Blots were analyzed with enhanced chemiluminescence reagents (Millipore) in a Chemidoc (Bio-rad). Beta-actin was used as a loading control. Uncropped blots are available in the Source Data File.

**Statistical analysis**. Statistical analyses was performed with GraphPad Prism 6.0 (GraphPad Software). Two-way analysis of variance (ANOVA), one-way ANOVA, *t*-test, and Mann–Whitney *U*-tests were used as required (specified in figure legends). In all analyses, statistical significance was set at $p < 0.05$. Unless otherwise stated, all error bars depict SEM. For ANOVA analysis, F is expressed as F (degrees of freedom, degree of freedom error) = x.

**In silico modeling of CLR01 pharmacokinetics**. Supplementary Methods detailing the in silico modeling of the pharmacokinetics of CLR01 are available. The code for the model is available upon request.

**Reporting summary**. Further information on experimental design is available in the Nature Research Reporting Summary linked to this paper.

## Data availability
Authors can confirm that all relevant data are included in the paper and/ or its supplementary information files. The code for the pharmacokinetics modeling in the brain is available upon request. Source data are available as a Source Data File. LR01 may be obtained from G.B. Source data are provided with this paper.

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

## Acknowledgements

We thank Sally Cowley, Jane Vowles, Phillippa Carling, and Marta Cherubini for their support with iPSCs and the manuscript. We thank the Network of Centres of Excellence in Neurodegeneration (COEN) MR/P007058/1, the Monument Trust Discovery Award from Parkinson's UK (J-1403), the National Institutes of Health (NIH/NIA R01AG050721), Team Parkinson/Parkinson Alliance, and the Medical Research Council Dementia Platform UK Stem Cell Network grant for support. The samples were obtained from the Brain Bank GIE NeuroCEB (BRIF number 0033-00011), funded by the patients' associations France Alzheimer, France Parkinson, ARSEP, and 'Connaître les Syndromes Cérébelleux' to which we express our gratitude.

## Author contributions

E.B., G.B., B.D., R.W.M., and C.M. conceived the project. N.B.V., R.W.M., E.B., G.B., B.D., F.C., and C.M. designed the experiments. N.B.V. performed experiments, analyzed the data, and prepared the first draft of the manuscript. B.D., E.F., P.R., S.V., I.S., and Z.L. performed experiments and analyzed the data. E.K., N.C.R., M.C., S.T., B.R., and S.C. provided expertise and assistance with FCV, animal work, and surgical procedures. L.V.B. created the in silico model of CLR01 pharmacology. G.B., F.K., and T.S. provided CLR01 and expert advice. R.W.M., E.B., C.M., F.C., B.D., S.C., and N.B.V. supervised the work. All authors contributed to the final version of the manuscript.

## Competing interests

The chemical composition of CLR01 is protected by International Patent No. PCT/US2010/026419, USA patent No. 8,791,092, and European patent No. EP2403859 A2.
