## [Peer Review File · Nature Communications]

Reviewers' comments:

Reviewer #1 (Remarks to the Author):

Based on publications by other groups showing that CLR01 is capable of inhibiting α -synuclein aggregation, this study by Bengoa-Vergniory and colleagues investigates the therapeutic potential of the small molecule CLR01 for Parkinson's disease. One major strength of this study is the strategic use of multiple complementary and translational experimental models of α -synuclein. The authors reported in this study that CLR01 attenuated protein aggregation, motor deficits and nigrostriatal damage. Overall, this is an interesting study with potential clinical relevance. The manuscript is well-written and data are clearly presented. However, following are some questions and comments:

Fig 1: The observation that both noLB and LB extracts induced a comparable extent of AK released to the medium by dead cells is rather unexpected. This result may indicate that cellular injury caused by these extracts was not mediated by α -synuclein. Also, please consult with a Biostatistician; it appears that one-way ANOVA is more appropriate than two-way ANOVA for this set of data analysis.

Fig 3, In addition to age-dependent changes in astrocytes and microglia, data should also be graphed together and analysed using two-way ANOVA to see if these alterations are genotype dependent.

Fig. 4. Experimental design for behavioral study (A-D) did not include the WT+CLR01 control group; therefore, it is not clear if the effect of CLR01 was specific to OVX mice. Additionally, the authors mention in the discussion section that at 12 months old these OVX mice do not have over dopaminergic neurodegeneration, indicating the observed motor deficits were not caused by nigrostriatal damage. Please discuss what might be responsible for these motor deficits.

Lastly, as a general comment, this study relies a bit too heavily on immunofluorescence for alterations in protein aggregation, astrocytes and microglia. Including a more objective quantitative method such as immunoblotting would strengthen these data.

Reviewer #2 (Remarks to the Author):

Review of "CLR01 protects dopaminergic neurons in vitro and in vivo in human neurons and mouse models of Parkinson's" (Bengoa-Verginiory and colleagues)

Summary: This paper describes a series of in vitro and in vivo evaluations of CLR01 in that demonstrate benefits of the compound on alpha-synuclein pathology and additional markers of brain health in Parkinson's disease-relevant model systems. The report is original and represents a substantial experimental effort. The readability and impact of the paper would be increased by a careful revision for clarity and flow. I would be happy to review a revised version of this manuscript.

Suggestions/Comments/Questions on specific content by section:

Abstract

- For readers who are unfamiliar with CLR01/molecular tweezers, I suggest introducing the class of CLR01 compounds (MOA) then referring to them as molecular tweezers.
- Simplify the description of in vivo experiments and shift description of SNCA-OVX model to the introduction or methods.

Introduction

- (Pg. 4, Paragraph 2) Incomplete sentence? "We have also shown previously that introducing the complete human SNCA locus on a mouse *Snca*^{-/-} background leads to spatial and temporal expression of human α -syn 12." --- ...that recapitulates the spatial and temporal expression of alpha-synuclein pathology in patients with PD? Corresponds to Braak-staging?

Methods

- Regarding Catwalk analyses – Were the individual trials evaluated for run effects? I understand that the data analyses can be onerous for Catwalk, however, it would be important to understand whether additional factors (e.g., fatigue or even response competition such as increased exploration during trials) play into the performance of the SNCA-OVX mice. Can you comment if you have already considered this possibility? If fatigue plays a role, then perhaps averaging 5 trials may occlude the motor phenotype. With regard to the recorded runs, do the animals look "different"? If so, are these differences captured by any of the Catwalk measures (even if subtle)?

- Regarding muscle strength evaluations – should consider subject weight in analysis of the inverted wire hang test (since the animal is not being pulled as in other grip assessments). Were these sessions videotaped? It would be interesting to know whether a greater fore or hind paw strength deficit could be teased out.

- Regarding use of TH as a marker for neuronal cell loss in SNc & striatum – While TH is often described as a marker for cell loss, it really should be considered a "neurochemical identity" marker. That is, the loss of TH does not necessarily mark cell loss/death. Suggest pairing the TH evaluation with cell counts in the SNc and DAT in the striatum (thereby strengthening evidence for SNc cell loss and translatable consequences in the striatum).

- No mention of fecal pellet evaluations – In what context was this evaluation made? Timed in a home cage? In the course of the locomotor evaluation? Have functional GI evaluations been conducted in the SNCA-OVX model? For future considerations, it would be very easy to conduct a timed food dye transport evaluation in these mice. Any indication of slowed whole gut transport would increase the value of this model for non-motor evaluations.

Results

- (Pg.5, 2nd Paragraph) "In agreement with this unique mechanism of action, CLR01 is well tolerated in multiple rodent models of neurodegeneration, and provides benefit on a wide range of pathological endpoints (e.g., alpha-synuclein, amyloid beta pathology) with a and has shown good tolerance and a high wide therapeutic dosing window in rodent models, which withstand high doses of this compound well 14. " As written, the original sentence does not support the assertion that the MOA of CLR01-like compounds has protein agnostic applications in a variety of proteinopathies.

- (Question) Do the authors have any concerns regarding EM specimen processing artifacts (i.e., changes in structure or assembling due to dehydration or other TEM specimen processing steps)?

- "Treatment with LB or noLB extracts produced different textures upon microscopic inspection;.." This is a bit unclear.... Does "different textures" refer to a morphological change or cellular distribution change of iPSCs? In the deposition of the protein? As visualized by α -syn immunofluorescence? What is the significance of the "texture" difference? Please clarify in text and also in Figure 1 legend.

- (Comment) The SNCA-OVX is described as having measurable motor dysfunction at 6, but not 4 months of age, despite measurable α -syn pathology starting at 3 months of age. It is surprising that the mice have no overt motor phenotype at 4 mo. The value of the model would be increased

by additional longitudinal motor-related evaluations (e.g., grip, nest building). While possibly outside the scope of the present manuscript, it may be useful to consider pharmacological assessments of underlying dopaminergic alterations that are present in these mice. Dose response evaluations of well-characterized dopaminergic interventions (e.g., relationship between amphetamine dose and locomotor activity levels) may unmask functional differences in dopaminergic neurotransmission in the SNCA-OVX mice.

- (Comment) I did not see any reference to/data for general in-life evaluations of animal health. Does this model have any changes in body weight or grooming relative to normal control mice? If so, is this age-dependent? And is there any indication that CLR01 has any beneficial effects on these endpoints? An opportunity to describe evaluations of fecal pellet output & content.

- (Pg. 11, Paragraph 1) RE: mention of LAMP2A and PGC1a markers and roles in CMA and mitophagy.

- o Please provide information regarding reagents and WB quantification (estimated MW for LAMP2A band? May provide important information for state of LAMP2A as an indicator of CMA deficits – glycosylated? Consult publications from Ana Marie-Cuervo's group for more information.

- o Were LAMP2A levels evaluated in age-matched control C57Bl/6 mice?

- o Suggest elaborating on significance of these two markers in normal aging and neurodegenerative disease patient populations in first few sentences of this section rather than after describing the changes.

- o How would you characterize the effects of CLR01 on these markers? As a direct effect or indirect effect (unburdening via lowering of oligomers)?

Discussion

- Can the authors comment on/compare CLR01/molecular tweezer approach to biological (antibody) interventions? In particular, there have been notable failures of the anti-a-syn antibodies in clinical trials. How might the use of CLR01 be a better approach?

Figures

- Figure 3 – Suggest including higher mag insets with examples of different glial morphology in presented images or thresholded/demarcated images?

- Figure 5 – include estimated molecular weights on Western Blot band images

- Suppl Figures 5 & 6 – (Comment) No mention of A-D findings (no clear phenotypic differences) in main text. Could mention this along with other in life observations in text.

- Suppl. Figure E – is this data for forefoot stride length or print length? Please clarify.

- Suppl. Figure 6 – comparing the data in Suppl. Figure 5H – looks like the overall levels of locomotor activity (K) are not further decreased at 18 months. Also, do you have the wildtype control data for these figures?

- Suppl. Figure 6 – At first glance, it almost seems like these is a reduced Catwalk motor phenotype at 18 months. Can this be explained by increased non-compliance or any other compensatory changes? Might be useful to evaluate how many animals were able to complete the walkway traversal in the allotted time or maybe did not even start traversal?

- Suppl. Figure 7 – Suggest adding higher mag inset panels for A & C to better illustrate changes.

- Suppl. Figure 8 – Lesions per SNC section = particles/puncta?

Reviewer #3 (Remarks to the Author):

This work asks whether CLR01 might be a potential therapeutic lead for synucleinopathies. It builds on interesting molecular studies, which had shown that CLR01 interacts reversibly with lysine/arginine residues to shield charges and limit synuclein aggregation in vitro (see ref 4-8). As a research tool, this is an interesting probe, which teaches us about molecular requirements.

The goals of this new work were to explore whether CLR01 might protect cells, tissues and ultimately animals from synuclein pathological effects. However, the work fails to convincingly achieve that goal, largely due to an incomplete treatment of the pharmacological studies. While there are some interesting ideas here, the pharmacological work contains too many gaps, assumptions and “wishful thinking” to be publishable.

1. In Figure 4 and 5, mice are treated with CLR01 (sub-Q or Alzet pump) and effects on synuclein phenotypes are studied. While some modest effects are seen, it is impossible to ascribe these to either the effects of CLR01 or its possible interaction with synuclein *in vivo*. A cornerstone of molecular pharmacology is that the potency (EC50) of a molecule must be determined (typically in cells or biochemical studies), and then the levels of the free compound in the tissue of interest (here, the CNS) is carefully measured to assure that the AUC/EC50 relationships make sense. In this manuscript, both the EC50 and the drug levels are unknown. Thus, one simply cannot conclude any pharmacological effect of the molecule on the animal. This is especially true for a molecule such as CLR01, which likely has thousands of targets *in vivo*, making the free drug concentration in CNS a critical factor. Importantly, one cannot simply point to literature here, one must account for mouse strain differences and perform all the PK/PD studies.

2. The switch from Figures 1-3 to Figures 4-5 seems to include a critical, new assumption that CLR01 gets into cells at an appreciable level to cause a pharmacological effect. It is unclear whether this assumption has a basis? Prior to the animal studies, synuclein or Lewy-Body derived material was pre-mixed with CLR01 prior to introduction to cells or tissues, invoking only an extracellular function. However, it would presumably need to enter cells and tissues (at high levels) to achieve a therapeutic effect. Frankly, this seems unlikely given the structure of the compound.

3. What is the utility of the flow chamber experiments in Figure 2? If CLR01 is reducing the content of synuclein oligomers in LB material (Fig 1), then it follows that it would decrease the amount that could be trafficked to other cells. For example, it seems that the same result could be achieved by simply decreasing the concentration of LB material that is added? What do we learn about CLR01 or its mechanism from these studies? The flow chambers are certainly a “fancy” platform (and quite useful for cellular mechanisms of synuclein trafficking), but here the result doesn’t add much.

4. The results in Figure 1 lack a number of critical controls. Firstly, the CLR01 experiments are performed at single concentrations. What is the dose dependence? Is the activity saturated? Is it at 50% of maximum? If activity is not dose dependent, it would suggest an alternative mechanism of non-specific cellular activation rather than target engagement. What is the relationship between compound dose and the total amount of LB material used? Second, what is the activity of a closely related analog that cannot bind lysines? Only PBS is used as a control, which doesn’t test the presence of high concentrations of an organic molecule.

Response to Reviewers: NCOMMS-19-14688A-Z

“CLR01 protects dopaminergic neurons in vitro and in vivo in human neurons and mouse model of Parkinson’s” by Bengoa-Vergniory et al

We would like to thank the Reviewers for the thorough review of our manuscript and the very positive comments made, including that **“One major strength of this study is the strategic use of multiple complementary and translational experimental models of α -synuclein”**, that **“Overall, this is an interesting study with potential clinical relevance”** and **“The report is original and represents a substantial experimental effort”**.

We have now added substantial new experimental data to address all points raised by the Reviewers. We have highlighted our all changes in the manuscript text in blue in order to facilitate the review process.

Reviewers' comments:

Reviewer #1 (Remarks to the Author):

Based on publications by other groups showing that CLR01 is capable of inhibiting α -synuclein aggregation, this study by Bengoa-Vergniory and colleagues investigates the therapeutic potential of the small molecule CLR01 for Parkinson’s disease. One major strength of this study is the strategic use of multiple complementary and translational experimental models of α -synuclein. The authors reported in this study that CLR01 attenuated protein aggregation, motor deficits and nigrostriatal damage. Overall, this is an interesting study with potential clinical relevance. The manuscript is well-written and data are clearly presented.

We thank the Reviewer for this summary, noting that **“One major strength of this study is the strategic use of multiple complementary and translational experimental models of α -synuclein”** and that **“Overall, this is an interesting study with potential clinical relevance. The manuscript is well-written and data are clearly presented.”**

However, following are some questions and comments:

Fig 1: The observation that both noLB and LB extracts induced a comparable extent of AK released to the medium by dead cells is rather unexpected. This result may indicate that cellular injury caused by these extracts was not be mediated by α -synuclein. Also, please consult with a Biostatistician; it appears that one-way ANOVA is more appropriate than two-way ANOVA for this set of data analysis.

We thank the Reviewer for this point. To investigate this further we have measured the contribution of the brain extract material itself to the adenylate kinase (AK) reading, before adding the extract to cultured neurons (see Reviewer Figure 1 below). We do obtain an AK reading solely from the brain extract itself, regardless of LB content, which is likely because these extracts will contain material from dead cells which generate the AK signal. As both noLB and LB extracts generate an AK reading even before adding to cells we have decided to remove the AK graphs (old manuscript Fig. 1D) from the study. We have also modified the new Figures in order to accommodate one-way ANOVA analyses for all groups and split the data into two different Figures (Fig. 1 and S4), which makes interpretation of the data easier.

Reviewer Figure 1: Adenylate kinase (AK) assay readings obtained from the noLB and LB brain extracts without addition to cultured cells. Although the AK reading is low, there is a contribution of the brain extract itself to the assay, regardless of LB content.

Fig 3, In addition to age-dependent changes in astrocytes and microglia, data should also be graphed together and analysed using two-way ANOVA to see if these alterations are genotype dependent.

We have amended the Figure as suggested.

Fig. 4. Experimental design for behavioral study (A-D) did not include the WT+CLR01 control group; therefore, it is not clear if the effect of CLR01 was specific to OVX mice. Additionally, the authors mention in the discussion section that at 12 months old these OVX mice do not have over dopaminergic neurodegeneration, indicating the observed motor deficits were not caused by nigrostriatal damage. Please discuss what might be responsible for these motor deficits.

We have not included a Wt+CLR01 group in this study for several reasons. First, we do not need to include such a group to investigate compound safety as the compound is well tolerated as described by Attar et al. (2014), who used higher doses than the ones used in this study in the same animal background, and therefore we would not expect any detrimental toxic effects. Second, such a study would add very little as wild-type control animals do not develop an a-syn accumulation pathology to clear. Therefore, in adherence with the good practice and the 3Rs policy in the UK of replacement, reduction and refinement for animal work, we did not think it ethical to use extra animals for a study which would be of little significant scientific value.

We thank the Reviewer for the second comment. We have rephrased the Discussion to say that “The window for treatment with a molecular tweezer is optimal at twelve months, when early disease processes are underway and neuronal dysfunction has begun, but before extensive and widespread dopaminergic neuron degeneration.” At this time point the cell pathology is subtle and not widespread.

Lastly, as a general comment, this study relies a bit too heavily on immunofluorescence for alterations in protein aggregation, astrocytes and microglia. Including a more objective quantitative method such as immunoblotting would strengthen these data.

We thank the Reviewer for this excellent suggestion and have now included further native immunoblotting analysis in order to validate our AS-PLA results at 18 months (Fig. 5H-I and S12E). This analysis shows that there are no SDS-insoluble aggregates in *SNCA-OVX* animals (as expected), that there is a modest accumulation of higher molecular weight aggregation in the SDS soluble fraction, which likely indicates the presence of a-syn oligomers, and that this accumulation is reduced with CLR01 treatment. This analysis also shows that most of the a-syn content in the midbrain of these animals is Triton soluble, which is consistent the idea that these are oligomeric aggregates. We have tried using WB techniques to assay the emergence of subtle cellular pathology at 12 months, but these crude total midbrain extracts fail to capture changes in specific cellular populations in the midbrain when changes are subtle.

We confirm that we performed all work, including the immunofluorescence, blind to genotype and treatment to ensure that analysis remained objective. We have added the native gel analysis of 18-month-old animals to Figure 5 and S12 and to the accompanying text. We have also included a sentence in the text acknowledging that we could not see the cellular phenotypes in 12-month-old animals by immunoblotting.

Reviewer #2 (Remarks to the Author):

Review of "CLR01 protects dopaminergic neurons in vitro and in vivo in human neurons and mouse models of Parkinson's" (Bengoa-Verginory and colleagues)

*Summary: This paper describes a series of in vitro and in vivo evaluations of CLR01 in that demonstrate benefits of the compound on alpha-synuclein pathology and additional markers of brain health in Parkinson's disease-relevant model systems. **The report is original and represents a substantial experimental effort.** The readability and impact of the paper would be increased by a careful revision for clarity and flow. I would be happy to review a revised version of this manuscript.*

We thank the Reviewer for this summary and note the comment that **"The report is original and represents a substantial experimental effort."**

Suggestions/Comments/Questions on specific content by section:

Abstract

•For readers who are unfamiliar with CLR01/molecular tweezers, I suggest introducing the class of CLR01 compounds (MOA) then referring to them as molecular tweezers.

We thank the Reviewer for this suggestion and have amended the manuscript accordingly. We have chosen to refer to CLR01 specifically in the Abstract due to word count constraints and because there are multiple other molecular tweezers and we wished to be precise.

•Simplify the description of in vivo experiments and shift description of SNCA-OVX model to the introduction or methods.

We thank the Reviewer for this suggestion and have amended the text accordingly.

Introduction

•(Pg. 4, Paragraph 2) Incomplete sentence? “We have also shown previously that introducing the complete human SNCA locus on a mouse *Snca*^{-/-} background leads to spatial and temporal expression of human α -syn 12.” --- ...that recapitulates the spatial and temporal expression of alpha-synuclein pathology in patients with PD? Corresponds to Braak-staging?

We thank the Reviewer for this suggestion and have amended the text accordingly.

Methods

• Regarding Catwalk analyses – Were the individual trials evaluated for run effects? I understand that the data analyses can be onerous for Catwalk, however, it would be important to understand whether additional factors (e.g., fatigue or even response competition such as increased exploration during trials) play into the performance of the SNCA-OVX mice. Can you comment if you have already considered this possibility? If fatigue plays a role, then perhaps averaging 5 trials may occlude the motor phenotype. With regard to the recorded runs, do the animals look “different”? If so, are these differences captured by any of the Catwalk measures (even if subtle)?

Reviewer Figure 2: Catwalk data presented as separate data for measurements on each day. The * indicates a difference ($p < 0.05$) between Wt and SNCA-OVX, and # indicates a difference between SNCA-OVX and SNCA-OVX treated with CLR01 ($p < 0.05$).

We have performed individual analysis of the each run per group as the Reviewer suggested (Reviewer Figure 2). It is clear that fatigue has no impact on our analysis because the animals actually ran *faster* in each consecutive run than in the previous up to Run 4, and then stabilizing. SNCA-OVX animals treated with CLR01 required more runs to perform a compliant run than Wts, and yet their speed was comparable, again suggesting no effect of fatigue. These individual analyses revealed no new differences, but we hope the analysis above clarifies this Reviewer comment.

•Regarding muscle strength evaluations – should consider subject weight in analysis of the inverted wire hang test (since the animal is not being pulled as in other grip assessments). Were these sessions videotaped? It would be interesting to know whether a greater fore or hind paw strength deficit could be teased out.

We thank the Reviewer for raising this important point. Our cohorts are all age, sex and weight matched, point we have now included in the text on description of the “Transgenic animal cohorts”. The muscle strength evaluation sessions were not videotaped.

•Regarding use of TH as a marker for neuronal cell loss in SNc & striatum – While TH is often described as a marker for cell loss, it really should be considered a “neurochemical identity” marker. That is, the loss of TH does not necessarily mark cell loss/death. Suggest pairing the TH evaluation with cell counts in the SNc and DAT in the striatum (thereby strengthening evidence for SNc cell loss and translatable consequences in the striatum).

We have performed these extra experiments requested by the Reviewer and they show changes in the same direction as the TH-positive neuron count data, although they do not reach significance. We have incorporated these new data in Supplementary Figure S13E-G and in the manuscript text.

•No mention of fecal pellet evaluations – In what context was this evaluation made? Timed in a home cage? In the course of the locomotor evaluation? Have functional GI evaluations been conducted in the SNCA-OVX model? For future considerations, it would be very easy to conduct a timed food dye transport evaluation in these mice. Any indication of slowed whole gut transport would increase the value of this model for non-motor evaluations.

We thank the Reviewer for this comment and we have amended the text accordingly, introducing the appropriate text into the Material and Methods section. Specifically, animals were placed into separate clean cages with no access to food or water, and fecal pellets were collected over a 1 hour period (from 1600–1700 hours). Pellets were weighed to obtain wet stool weight, dried overnight at 65°C, and reweighed to obtain dry stool weight and stool water content. We have not conducted any functional GI evaluations in the SNCA-OVX model, such as to conduct a timed food dye transport evaluation.

Results

•(Pg.5, 2nd Paragraph) “In agreement with this unique mechanism of action, CLR01 is well tolerated in multiple rodent models of neurodegeneration, and provides benefit on a wide range of pathological endpoints (e.g., alpha-synuclein, amyloid beta pathology) with a and has shown good tolerance and a high wide therapeutic dosing window in rodent models, which withstand high doses of this compound well 14. “ As written, the original sentence does not support the assertion that the MOA of CLR01-like compounds has protein agnostic applications in a variety of proteinopathies.

We thank the Reviewer for this comment and have amended the text to “CLR01 binds specifically with high on-off rate to lysines and, therefore, its specificity is not to a particular protein, but to the process of abnormal protein self-assembly itself. CLR01 has shown good tolerance and a high therapeutic dosing window in rodent models, which withstand high doses of this compound well”.

•(Question) Do the authors have any concerns regarding EM specimen processing artifacts (i.e., changes in structure or assembling due to dehydration or other TEM specimen processing steps)?

Given that our EM processing and staining method is one widely used in the literature, and that our standard aggregation protocol matches that of others, we do not expect any significant artefacts. Our negative-staining EM protocol uses aqueous solutions and it is therefore unlikely that there would be any dehydration, besides the final drying of the grid. It is also highly unlikely that any of the other steps in the protocol (2 min contact with the grid, a 10 second 2% uranyl acetate stain, and 1 second water rinse) would affect the structure of these aggregates.

•“Treatment with LB or noLB extracts produced different textures upon microscopic inspection;..” This is a bit unclear.... Does “different textures” refer to a morphological change or cellular distribution change of iPSCs? In the deposition of the protein? As visualized by a-syn immunofluorescence? What is the significance of the “texture” difference? Please clarify in text and also in Figure 1 legend.

We thank the Reviewer for this comment and have now clarified the wording. We used the term “texture” to refer to the appearance of aggregated material from the brain extract deposited on the outside surface of the cells.

•(Comment) The SNCA-OVX is described as having measurable motor dysfunction at 6, but not 4 months of age, despite measurable a-syn pathology starting at 3 months of age. It is surprising that the mice have no overt motor phenotype at 4 mo. The value of the model would be increased by additional longitudinal motor-related evaluations (e.g., grip, nest building). While possibly outside the scope of the present manuscript, it may be useful to consider pharmacological assessments of underlying dopaminergic alterations that are present in these mice. Dose response evaluations of well-characterized dopaminergic interventions (e.g., relationship between amphetamine dose and locomotor activity levels) may unmask functional differences in dopaminergic neurotransmission in the SNCA-OVX mice.

We must correct the Reviewer on this point and clarify that we do **not** expect a motor phenotype at 6 months. The earliest time point we have observed motor phenotypes is at 12 months, which we hope is clear from this amended text “As an end-point for our 12- and 18-month cohorts, we assessed the animals for both motor behavior and neuropathology, whereas the 6-month-old cohort was evaluated for dopamine release and microglial morphology. The 6-month cohort would be too young to show a neuronal-loss-associated motor phenotype, considering our previous findings.”

We have no evidence that there is an accumulation of a-syn pathology at 3 months in SNCA-OVX animals compared to Wt animals. Rather, a-syn oligomeric deposition starts increasing at later time points from 6-9 months. We agree that pharmacological assessments of underlying dopaminergic alterations present in the SNCA-OVX mice, such as dose response evaluations of well-characterized dopaminergic interventions (for example, using amphetamine dosing), are beyond the scope of the current study.

•(Comment) I did not see any reference to/data for general in-life evaluations of animal health. Does this model have any changes in body weight or grooming relative to normal control mice? If so, is this age-dependent? And is there any indication that CLR01 has any beneficial effects on these endpoints? An opportunity to describe evaluations of fecal pellet output & content.

We did not collect any grooming or other general in-life evaluation data of animal health in Wt and SNCA-OVX animals, although the SNCA-OVX seem more placid when handled. We did not classify stool by appearance (or according to the Bristol stool chart) before the overnight drying step, which then precluded any subsequent evaluation, but we will consider it in future investigations.

•(Pg. 11, Paragraph 1) RE: mention of LAMP2A and PGC1a markers and roles in CMA and mitophagy.
o Please provide information regarding reagents and WB quantification (estimated MW for LAMP2A band? May provide important information for state of LAMP2A as an indicator of CMA deficits – glycosylated? Consult publications from Ana Marie-Cuervo’s group for more information.
o Were LAMP2A levels evaluated in age-matched control C57Bl/6 mice?
o Suggest elaborating on significance of these two markers in normal aging and neurodegenerative disease patient populations in first few sentences of this section rather than after describing the changes.
o How would you characterize the effects of CLR01 on these markers? As a direct effect or indirect effect (unburdening via lowering of oligomers)?

We thank the Reviewer for this important comment and we have clarified the text accordingly. We have also provided molecular weights on the side of WBs. We did not have a Wt control group at 18 months, however, we have now elaborated on the significance of Lamp2a in PD and have suggested that the effects might indeed be an indirect effect of aggregate dissociation.

Discussion

- *Can the authors comment on/compare CLR01/molecular tweezer approach to biological (antibody) interventions? In particular, there have been notable failures of the anti-a-syn antibodies in clinical trials. How might the use of CLR01 be a better approach?*

We thank the Reviewer for this comment and have amended the text accordingly. Based on the new data we present in response to Reviewer 3 below, we have introduced a comment on the ability of CLR01 to penetrate into cells which represents an advantage to other strategies, such as the current anti-a-syn antibody therapies.

Figures

- *Figure 3 – Suggest including higher mag insets with examples of different glial morphology in presented images or thresholded/demarcated images?*

We thank the Reviewer for this comment and we have added a new Figure (new Fig. S7) where we show these images.

- *Figure 5 – include estimated molecular weights on Western Blot band images*

We have amended the Figure accordingly.

- *Suppl Figures 5 & 6 – (Comment) No mention of A-D findings (no clear phenotypic differences) in main text. Could mention this along with other in life observations in text.*

We thank the Reviewer for this comment and have amended the text accordingly.

- *Suppl. Figure E – is this data for forefoot stride length or print length? Please clarify.*

We thank the Reviewer for this comment and have amended the text to clarify that the data are stride length.

- *Suppl. Figure 6 – comparing the data in Suppl. Figure 5H – looks like the overall levels of locomotor activity (K) are not further decreased at 18 months. Also, do you have the wildtype control data for these figures?*

We do not have a Wt cohort for this experiment. Although we keep all variables as similar as possible across the cohorts for the behavioural analysis (time of day, cages, bedding, light, for example), we cannot control every variable that could potentially affect the animals (noise, vibrations, etc.) We therefore restrict our comparisons to those made within cohorts that have gone through the same experience together at the same time. Also, there is only 4 months of difference between animals treated for two months from 12 months (ie: aged 14 months at the time of the behavioural assay) and the 18-month-old animals, and so differences may be not detectable.

- *Suppl. Figure 6 – At first glance, it almost seems like these is a reduced Catwalk motor phenotype at 18 months. Can this be explained by increased non-compliance or any other compensatory changes? Might be useful to evaluate how many animals were able to complete the walkway traversal in the allotted time or maybe did not even start traversal?*

At the 18 months timepoint the average of 5 compliant runs was taken forward for analysis, consistent with the approach taken at 12 months. As with the 12 month data, we did not observe any differences in compliance at 18 months. While there was a modest decrease in some of the parameters, it did not reach statistical significance.

At 18 months of age dopaminergic neurons are already lost in the *SNCA-OVX* model so the objective of this late-stage cohort was focused on neuropathological analysis rather than behavioural analysis. As the dead neurons cannot be recovered we did not expect behavioural improvements at this timepoint and therefore n numbers were lower for the 18-month neuropathological cohort than for the 12 month behavioural cohort (7/8 versus 11/14). We would not, therefore, wish to over interpret non-significant results from the low-numbers in the 18-month-old behaviour data.

- *Suppl. Figure 7 – Suggest adding higher mag inset panels for A & C to better illustrate changes.*

We have amended the Supplementary Figure as requested.

- *Suppl. Figure 8 – Lesions per SNC section = particles/puncta?*

We have clarified the Supplementary Figure legend as requested.

Reviewer #3 (Remarks to the Author):

This work asks whether CLR01 might be a potential therapeutic lead for synucleinopathies. It builds on interesting molecular studies, which had shown that CLR01 interacts reversibly with lysine/arginine residues to shield charges and limit synuclein aggregation in vitro (see ref 4-8). As a research tool, this is an interesting probe, which teaches us about molecular requirements. The goals of this new work were to explore whether CLR01 might protect cells, tissues and ultimately animals from synuclein pathological effects. However, the work fails to convincingly achieve that goal, largely due to an incomplete treatment of the pharmacological studies. While there are some interesting ideas here, the pharmacological work contains too many gaps, assumptions and “wishful thinking” to be publishable.

We have incorporated substantial new experimental data to address all the points raised by the Reviewer. We have added major new experimental data to demonstrate that the drug is able penetrate cells, that the drug has good pharmacological properties, including a nanomolar EC50, and have added additional work with the inactive counterpart CLR03 as requested.

1. In Figure 4 and 5, mice are treated with CLR01 (sub-Q or Alzet pump) and effects on synuclein phenotypes are studied. While some modest effects are seen, it is impossible to ascribe these to either the effects of CLR01 or its possible interaction with synuclein *in vivo*. A cornerstone of molecular pharmacology is that the potency (EC50) of a molecule must be determined (typically in cells or biochemical studies), and then the levels of the free compound in the tissue of interest (here, the CNS) is carefully measured to assure that the AUC/EC50 relationships make sense. In this manuscript, both the EC50 and the drug levels are unknown. Thus, one simply cannot conclude any pharmacological effect of the molecule on the animal. This is especially true for a molecule such as CLR01, which likely has thousands of targets *in vivo*, making the free drug concentration in CNS a critical factor. Importantly, one cannot simply point to literature here, one must account for mouse strain differences and perform all the PK/PD studies.

We thank the Reviewer for raising this important point and we have now undertaken additional experiments in order to address the questions raised on the pharmacology of the compound. We have transfected SH-SY5Y cells with an a-syn expression plasmid and used the a-syn PLA assay to detect only intracellular a-syn oligomers. We found that treatment with increasing concentrations of CLR01 significantly reduces the a-syn oligomeric content of these cells with an *in vitro* EC50 of 85 nM, indicating a high level of potency (Fig. S3).

We agree with the further point that mouse strain differences are critical when evaluating the pharmacology of a compound. Fortunately, this information exists in the literature for the mouse strain (C57Bl/6) we have used for all the work in our manuscript. In previous work Attar et al. (2014) have already showed that CLR01 is able to achieve levels of 2-8 µg/ml maintained for 5 h after one subcutaneous drug dose, and that 1-3% of the blood-available drug reached the CNS. The same study also showed that repeated injections led to an accumulation of the drug in the brain. Based on their pharmacology data, CLR01 should get into the CNS of our treated animals at the level of 3-41 nM per injection. Considering that animals get a total of 16 injections (or the equivalent of 8 injections when using mini-pumps), we are confident that the drug concentration will reach the 85 nM EC50 calculated *in vitro* for α-syn oligomer dissociation. We have made appropriate comments in the text referring to this important information.

2. The switch from Figures 1-3 to Figures 4-5 seems to include a critical, new assumption that CLR01 gets into cells at an appreciable level to cause a pharmacological effect. It is unclear whether this assumption has a basis? Prior to the animal studies, synuclein or Lewy-Body derived material was pre-mixed with CLR01 prior to introduction to cells or tissues, invoking only an extracellular function. However, it would presumably need to enter cells and tissues (at high levels) to achieve a therapeutic effect. Frankly, this seems unlikely given the structure of the compound.

The Reviewer raises an important aspect of the paper which we have now addressed. In data published only very recently and since we submitted the original version of the manuscript, our collaborators have studied the internalization of CLR01 into cells (Herrera-Vaquero et al. 2019). By labelling CLR01 with a fluorescent tag the authors were able to confirm that CLR01 is taken up into cells. Adding further to these results, we have developed the assay as described above in which we evaluate the ability of CLR01 to dissociate intracellular α-syn oligomers (Fig. S3). We first transfected SH-SY5Y cells with a plasmid expressing α-syn and then measured oligomeric α-syn upon CLR01 treatment using the a-syn PLA assay. We found that CLR01 applied to the outside of the cells significantly reduces intracellular α-syn oligomers *in vitro* 48 hours after treatment.

3. What is the utility of the flow chamber experiments in Figure 2? If CLR01 is reducing the content of synuclein oligomers in LB material (Fig 1), then it follows that it would decrease the amount that could be trafficked to other cells. For example, it seem that the same result could be achieved by simply decreasing the concentration of LB material that is added? What do we learn about CLR01 or its mechanism from these studies? The flow chambers are certainly a “fancy” platform (and quite useful for cellular mechanisms of synuclein trafficking), but here the result doesn’t add much.

The utility of the flow chamber experiments is to demonstrate, for the first time in iPSC-derived dopamine neurons, that oligomers can be actively transported along axons and, more precisely, that they can be transported retrogradely. In addition to this novel finding indicating that axonal transport of a-syn occurs, we have shown that CLR01 inhibits this process. We have also added to the revised manuscript further experiments showing that a-syn is able to interact both with dynein and kinesin (the retro- and anterograde transport proteins, respectively) and that CLR01 is able to specifically reduce interactions between a-syn and these transport proteins. That is also an entirely novel result and highlights that CLR01 affects axonal transport of a-syn as well as aggregation (Fig. S6).

4. The results in Figure 1 lack a number of critical controls. Firstly, the CLR01 experiments are performed at single concentrations. What is the dose dependence? Is the activity saturated? Is it at 50% of maximum? If activity is not dose dependent, it would suggest an alternative mechanism of non-specific cellular activation rather than target engagement. What is the relationship between compound dose and the total amount of LB material used? Second, what is the activity of a closely related analog that cannot bind lysines? Only PBS is used as a control, which doesn’t test the presence of high concentrations of an organic molecule.

We thank the Reviewer for highlighting these points which we have now addressed. We have added new experimental data showing the dose-dependence of the compounds in vitro (se Fig. S3 and 5). CLR01 activity is saturated at 10 μ M, the dose widely used in the literature and the dose we used for our work in vitro. Working with brain extract, cells were treated with 600 pg/ml of LB/noLB extract. We decided to use this concentration of extract based on our pilot experiments which showed that there was a reduction in TH+ stained area in wells treated with this level of LB extracts (Reviewer Figure 3), which is also in agreement with the degeneration of neuronal processes we observe.

Reviewer Figure 3: TH+ stained area was averaged for 9 independent fields and compared to other treatments in iPSC-derived dopaminergic cultures.

In our work we decided to use PBS vehicle as a control rather than an inactive analogue because the analogue used widely in the literature (CLR03) has been previously shown to promote oligomerization of Abeta42 (see Zheng et al. 2015). We therefore did not wish to confound our work by promoting a possible alternative pathology in the control groups in vivo.

In response to the Reviewer's comment we have now generated new experimental data and provide additional controls for both the intracellular and extracellular dissociation of a-syn aggregates in Fig. S3 and 5 showing that the inactive analogue CLR03 does not affect a-syn dissociation.

References:

Attar et al, BMC Pharmacology and Toxicology 2014, 15:23

Herrera-Vaquero et al, Molecular basis of disease 2019, 165513

Zheng et al, Journal of Physical Chemistry 2015, 119, 14

Reviewers' comments:

Reviewer #1 (Remarks to the Author):

In this revised manuscript the authors have addressed most of the concerns raised by this reviewer. However, they did not provide satisfactory responses to the concerns in regards to motor function assessment in Fig. 4A-D. First, regarding the control group (WT+CLR01), the concern was whether CLR01 itself would have stimulatory effects (as seen with methamphetamine) on animal motor function. The original study reported by Attar et al. did not characterise the effects of CLR01 on animal locomotor function. Although this reviewer appreciates the practice of 3Rs policy, failing to include an appropriate control group does not serve the purpose well. Second, the authors still did not provide an explanation as to why these animals exhibited such a dramatic motor function impairment in the absence of nigrostriatal degeneration. In patients with Parkinson's, unless the loss of striatal dopamine content reaches the threshold of more than 80%, abnormal movements do not occur. It is possible that such observations in these animals were caused by peripheral effects, not central. The authors are recommended to consider removing this set of data.

Reviewer #2 (Remarks to the Author):

The authors thoughtful consideration of and response to all Reviewer comments in the revised manuscript are clear and well done. The revisions include the inclusion of additional data, which is appreciated as a considerable effort. This reviewer has no remaining issues with the revised manuscript.

Reviewer #3 (Remarks to the Author):

The author have made a number of improvements in the manuscript and should be commended. Given the new results, this reviewer would support publication – but some additional attention should be made around two points.

1. The description of the CLR01 animal pharmacology requires additional clarity. The authors now report the cellular potency of CLR01 as $EC_{50} \sim 85$ nM, which is important. As an exercise, we can use the reported level of CLR01 in mouse CSF, which is measured at around 3 to 41 nM for 5 hours. The free fraction (e.g. non-protein bound) of CLR01 is not known, but let us assume that this value is around the average of ~ 10 to 50%. So, even if 8 injections lead to a linear increase in C_{max} and AUC (which is unlikely/impossible, given the contribution of metabolism and clearance rates), a rough calculation yields an upper limit of 12 to 320 nM CLR01 during a portion of the experiment – which is a value that is either significantly below the EC_{50} or perhaps above it. The only point of this brief, semi-quantitative discussion is to emphasize that the project requires a more rigorous treatment of drug pharmacology. At present, it is still difficult to judge whether the target is engaged or to what extent, as the dosing scheme is not adequately justified. The addition of a table with the AUC, C_{max} , and $t_{1/2}$ values for CLR01 would be quite helpful. How long is the compound above EC_{50} in the brain? This reviewer suggests that the Editor and Author might identify a highly trained CNS drug pharmacologist, who can mathematically model the results and determine a rigorous target engagement value. Such changes would further increase the clarity of the work and allow for better replication and evaluation.
2. The membrane permeability of CLR01 remains unclear. The gold-standard experiment for permeability is to use PAMPA or similar assays, especially MDCK and CaCo. These experiments are readily done at contract research organizations (CROs) and can be informative in predicting/modeling permeability to determine cytosolic partitioning. At present, the cellular data on CLR01 could be explained by the compound's binding to a surface target, such as GPCR or ion channel, which mediates signaling effects in the cytosol (ultimately effecting the synuclein biomarker). To be clear, this mechanistic possibility is also potentially interesting and the point of

this comment is not to be negative. Rather, care must be taken in assuming permeability, especially for anionic molecules (which are rarely passively permeable and only occasionally trafficked). This reviewer recommended that the authors be transparent in the Discussion about the challenges with CLR01, including its uncertain permeability and its putative off-target effects. A more balanced tone seems warranted.

Response to Reviewers: NCOMMS-19-14688B-Z

“CLR01 protects dopaminergic neurons in vitro and in vivo in human neurons and mouse models of Parkinson’s” by Bengoa-Vergniory et al

We note the Reviewers’ very positive comments that **“In this revised manuscript the authors have addressed most of the concerns raised by this reviewer”**, that **“The authors thoughtful consideration of and response to all Reviewer comments in the revised manuscript are clear and well done. The revisions include the inclusion of additional data, which is appreciated as a considerable effort. This reviewer has no remaining issues with the revised manuscript.”** and **“The authors have made a number of improvements in the manuscript and should be commended. Given the new results, this reviewer would support publication...”**

We have added new experimental data to address all points raised by the Reviewers as detailed below in our point-by-point rebuttal. We have highlighted our all changes in the manuscript text in blue to facilitate the review process.

Reviewers' comments:

Reviewer #1 (Remarks to the Author):

In this revised manuscript the authors have addressed most of the concerns raised by this reviewer. However, they did not provide satisfactory responses to the concerns in regards to motor function assessment in Fig. 4A-D. First, regarding the control group (WT+CLR01), the concern was whether CLR01 itself would have stimulatory effects (as seen with methamphetamine) on animal motor function. The original study reported by Attar et al. did not characterise the effects of CLR01 on animal locomotor function. Although this reviewer appreciates the practice of 3Rs policy, failing to include an appropriate control group does not serve the purpose well.

We thank the Reviewer for this summary, noting that **“In this revised manuscript the authors have addressed most of the concerns raised by this reviewer”**.

At the Reviewer’s request we have now dosed a further small cohort of Wt and SNCA-OVX animals with the same dosing paradigm of CLR01 and subjected animals to rotarod and catwalk analysis. Wt animals showed no changes in their behavior after CLR01 treatment as measured by both rotarod and catwalk, whereas mild improvement was seen in the SNCA-OVX animals, suggesting that CLR01 may require the presence of aggregates in order to elicit its effects. Therefore, we can conclude that CLR01 itself has no stimulatory effects on animal motor function. We have added the new information into the manuscript and in Supplementary Figure 12.

We also wish to highlight that if CLR01 was indeed stimulatory then we would observe stimulatory effects, as suggested by the Reviewer in locomotor analysis, which we do not observe (Fig. 11H). All this evidence suggests that CLR01 effects are not driven by unspecific stimulatory effects, but rather by effects that are specific to SNCA-OVX animals, and likely due to the dissociation of aggregates.

Second, the authors still did not provide an explanation as to why these animals exhibited such a dramatic motor function impairment in the absence of nigrostriatal degeneration. In patients with Parkinson’s, unless the loss of striatal dopamine content reaches the threshold of more than 80%, abnormal movements do not occur. It is possible that such observations in these animals were caused by peripheral effects, not central. The authors are recommended to consider removing this set of data.

In response to this comment, we must point to the large and increasing amount of recent evidence from a range of transgenic rodent models of Parkinson’s which consistently show that a modest loss

of dopaminergic neurons (or in some cases even no loss) **elicits motor defects due to dopamine neuron dysfunction preceding neuronal death** in several independent transgenic alpha-synuclein (Janezic et al. 2013, Tofaris et al. 2006, Campos et al. 2013), and *LRRK2* models (Sloan et al. 2016, Walker et al. 2014, Tong et al. 2009), and in toxin-based parkinsonian models (Goldberg et al. 2011, Munoz-Machado et al. 2015). Specifically:

- Janezic et al. reported a 30% SNc dopamine neuron loss, a 30% dopaminergic release deficit specific to the dorsal striatum with a 40% reduction in latency to fall in rotarod, in the *SNCA-OVX* model.
- Tofaris et al. report that expression of human truncated alpha-synuclein was able to reduce dopamine by 30% in the striatum and cause a 20% reduction of locomotor activity in the absence of cell loss.
- Campos et al. showed that AAV-*SNCA* injections into the SNc elicited at 30% cell loss and a 30% reduction in latency to fall by rotarod.
- Sloan et al. showed that in the absence of neurodegeneration a 40-60% decrease in latency to fall by rotarod was observed in two transgenic *LRRK2* rat models (*LRRK2-G2019S* and *LRRK2-R1441C*) which was then corrected by L-DOPA treatment.
- Walker et al. demonstrated in a *LRRK2* rat overexpression transgenic model that behavioural effects could be detected in the absence of dopaminergic cell loss.
- Tong et al. reported decreased locomotor activity response to the inhibitory effect of a D2 receptor agonist, quinpirole, in a *Lrrk2* R1441C knockin mouse model.
- Goldberg et al. show several motor phenotypes with a SNc dopaminergic cell loss of 20% and a striatal TH+ denervation of 30%, including rearing and locomotor activity in an MPTP model.
- Munoz-Machado et al. report manual catwalk analysis with decreased forelimb strides of roughly 15 and 25% with a SNc cell loss of 25 and 50% and dopamine content reduction of 20 and 50%, respectively, in an MPTP mouse model.

Overall, we propose that the reduction in motor activity we observe in the absence of frank neuronal death in our model is consistent with the decreased dopamine neurotransmission we have reported (Janezic et al 2013), supported by the literature cited above.

References

- Janezic et al. 2013 <https://doi.org/10.1073/pnas.1309143110>
- Tofaris et al. 2006 <https://doi.org/10.1523/JNEUROSCI.4965-05.2006>
- Campos et al. 2013 <https://doi.org/10.3389/fnbeh.2013.00175>
- Sloan et al. 2016 <https://www.ncbi.nlm.nih.gov/pmc/articles/PMC4754049/>
- Walker et al. 2014 <https://www.ncbi.nlm.nih.gov/pubmed/25000966>
- Tong et al. 2009 <https://www.pnas.org/content/106/34/14622.short>
- Goldberg et al. 2011 <https://doi.org/10.1016/j.neuroscience.2011.02.027>
- Munoz-Machado et al. 2013 <https://doi.org/10.1111/jnc.13409>

Reviewer #2 (Remarks to the Author):

The authors thoughtful consideration of and response to all Reviewer comments in the revised manuscript are clear and well done. The revisions include the inclusion of additional data, which is appreciated as a considerable effort. This reviewer has no remaining issues with the revised manuscript.

We thank the Reviewer for this summary, noting that **“The authors thoughtful consideration of and response to all Reviewer comments in the revised manuscript are clear and well done. The revisions include the inclusion of additional data, which is appreciated as a considerable effort.”**.

Reviewer #3 (Remarks to the Author):

The author have made a number of improvements in the manuscript and should be commended. Given the new results, this reviewer would support publication – but some additional attention should be made around two points.

We thank the Reviewer for this response, noting that **“The authors have made a number of improvements in the manuscript and should be commended. Given the new results, this reviewer would support publication...”**

1. The description of the CLR01 animal pharmacology requires additional clarity. The authors now report the cellular potency of CLR01 as EC50 ~ 85 nM, which is important. As an exercise, we can use the reported level of CLR01 in mouse CSF, which is measured at around 3 to 41 nM for 5 hours. The free fraction (e.g. non-protein bound) of CLR01 is not known, but let us assume that this value is around the average of ~10 to 50%. So, even if 8 injections lead to a linear increase in Cmax and AUC (which is unlikely/impossible, given the contribution of metabolism and clearance rates), a rough calculation yields an upper limit of 12 to 320 nM CLR01 during a portion of the experiment – which is a value that is either significantly below the EC50 or perhaps above it. The only point of this brief, semi-quantitative discussion is to emphasize that the project requires a more rigorous treatment of drug pharmacology. At present, it is still difficult to judge whether the target is engaged or to what extent, as the dosing scheme is not adequately justified. The addition of a table with the AUC, Cmax, and t1/2 values for CLR01 would be quite helpful. How long is the compound above EC50 in the brain? This reviewer suggests that the Editor and Author might identify a highly trained CNS drug pharmacologist, who can mathematically model the results and determine a rigorous target engagement value. Such changes would further increase the clarity of the work and allow for better replication and evaluation.

We thank the Reviewer for this useful and constructive comment. We have followed their suggestion and built an *in silico* model with a specialist in mathematical modelling (Brown et al. 2018a, Brown et al. 2018b, Saeed-Vafa et al. bioRxiv) to determine AUC, Cmax and time over EC50, taking into account our dosing scheme. We have also performed blood brain barrier permeability experiments, specific to animals of the correct age and genotype and using the same dosing paradigm to take all relevant factors into account.

Blood brain barrier experiments showed that 10-35% of the concentration CLR01 in the blood crossed into the brain, and that this is a dose and time-dependent process. Based on this data, and previously published blood kinetics (Attar et al. 2014), we firstly predicted the extra and intracellular accumulation of the compound in brain (Fig. S8B), and secondly we simulated a 2 month treatment

spread across 16 sub-cutaneous injections at 40 µg/kg/day (Fig. S8C) in order to obtain pharmacokinetic parameters (Sup. Table 1).

We have found that CLR01 shows favourable pharmacokinetics and is above EC₅₀ levels (69.5 ng/ml) 93% of the time in the intracellular compartment (Fig. S8 and Sup. Table 1), with a time averaged AUC of 240 ng/ml and a C_{max} of 510 ng/ml. Taken together, these new data suggest the brain and cell pharmacology are above EC₅₀ levels, and therefore reinforce the importance of our findings and support the potential of CLR01 as a therapeutic drug.

2. The membrane permeability of CLR01 remains unclear. The gold-standard experiment for permeability is to use PAMPAs or similar assays, especially MDCK and CaCo. These experiments are readily done at contract research organizations (CROs) and can be informative in predicting/modeling permeability to determine cytosolic partitioning. At present, the cellular data on CLR01 could be explained by the compound's binding to a surface target, such as GPCR or ion channel, which mediates signaling effects in the cytosol (ultimately effecting the synuclein biomarker). To be clear, this mechanistic possibility is also potentially interesting and the point of this comment is not to be negative. Rather, care must be taken in assuming permeability, especially for anionic molecules (which are rarely passively permeable and only occasionally trafficked). This reviewer recommended that the authors be transparent in the Discussion about the challenges with CLR01, including its uncertain permeability and its putative off-target effects. A more balanced tone seems warranted.

We thank the Reviewer for this comment. We have conducted additional characterization of the uptake of CLR01 by using TAMRA-labelled CLR01, which is part of a different manuscript in preparation but we have added the key data to this response for the benefit of the Reviewers and rebuttal purposes (Reviewer Figure 1). As is clear from Reviewer Figure 1A, CLR01 (red) does not co-localize with mitotracker or p62, indicating that CLR01 does not accumulate in mitochondria or autophagosomes. However, CLR01 co-localized with lysotracker and Rab7a, which indicates that CLR01 is found in lysosomes and endosomes. When we followed this process as a time course (Reviewer Figure 1C-D), we could observe that uptake of CLR01 to endo-lysosomal compartments was time-dependent. We have therefore shown, that CLR01 is taken up through the endo-lysosomal system into the cell, which is very likely how this anionic compound crosses the plasma membrane.

Reviewer Figure 1. CLR01 enters the cell via the endo-lysosomal pathway. TAMRA-labelled CLR01 (red) was applied at 5 μ M to the medium of SH-SY5Y cells which were co-labelled after 24 hours with (A, left panel) Mitotracker (cyan); (B, left panel) Lysotracker (green); (A, right panel) p62 (green) or (B, right panel) by co-expressing Rab7a-GFP. (C) Actin-GFP expressing cells were labelled with Lysotracker (cyan) and used to evaluate the lysosomal compartment as a time-course (30 min – 18 h) after 5 μ M TAMRA-CLR01 treatment. (D) Rab7a-GFP expressing cells were used to evaluate the endosomal compartment as a time-course of 30 min – 18 h after 5 μ M TAMRA-CLR01 treatment.

References

- Attar et al. 2014 <https://www.ncbi.nlm.nih.gov/pubmed/24735982>
- Brown et al. 2018a <https://onlinelibrary.wiley.com/doi/full/10.1111/cei.13182>
- Brown et al. 2018b <https://royalsocietypublishing.org/doi/10.1098/rsif.2018.0041>
- Saeed-Vafa et al. bioRxiv <https://www.biorxiv.org/content/10.1101/190561v1>

REVIEWERS' COMMENTS:

Reviewer #1 (Remarks to the Author):

In this resubmission the authors have addressed largely this reviewer's concerns. It is appreciated that additional locomotor experiments were performed. However, interpretation of these new data (Supplementary Figure 12) should be revised in the manuscript. The conclusion that "...mild improvement was seen in the SNCA-OVX animals, suggesting that CLR01 may require the presence of aggregates in order to elicit its effects" is not supported by the data – unless statistical significance is denoted. Accordingly, a brief discussion should be provided regarding why these new data do not reproduce the effects observed in Figure 4 (perhaps due to a small sample size?). Without such clarification, these conflicting data in the SNCA-OVX mice will generate questions from the readers.

Reviewer #3 (Remarks to the Author):

The more rigorous treatment of pharmacokinetics is a significant improvement, allowing conclusions to be made about target engagement. To be clear, membrane permeability is still inconclusive (the fluorophore experiment provided in the response document actually suggests that the CLR01 molecule is not able to traverse the membrane and, rather, gets stuck in the endosome). However, on balance, the manuscript is significantly improved and publication is recommended.

Response to Reviewers: NCOMMS-19-14688B-Z
“CLR01 protects dopaminergic neurons in vitro and in vivo in human neurons and mouse models of Parkinson’s” by Bengoa-Vergniory et al

We thank the Reviewers for their very positive comments on our revised manuscript. The Reviewers note that **“the authors have addressed largely this reviewer’s concerns”**, that **“It is appreciated that additional locomotor experiments were performed”** and that **“the manuscript is significantly improved and publication is recommended”**. We are delighted that the work has been accepted in principle for publication in *Nature Communications*.

Reviewer 1

In this resubmission the authors have addressed largely this reviewer’s concerns. It is appreciated that additional locomotor experiments were performed. However, interpretation of these new data (Supplementary Figure 12) should be revised in the manuscript. The conclusion that “...mild improvement was seen in the SNCA-OVX animals, suggesting that CLR01 may require the presence of aggregates in order to elicit its effects” is not supported by the data – unless statistical significance is denoted. Accordingly, a brief discussion should be provided regarding why these new data do not reproduce the effects observed in Figure 4 (perhaps due to a small sample size?). Without such clarification, these conflicting data in the SNCA-OVX mice will generate questions from the readers.

We thank the Reviewer for the appreciative comments on the new locomotor data experiments we performed. As requested, we have added a brief comment (underlined) to the manuscript noting that the smaller sample size in this extra dataset may account for the lack of statistical significance observed.

Reviewer 3

The more rigorous treatment of pharmacokinetics is a significant improvement, allowing conclusions to be made about target engagement. To be clear, membrane permeability is still inconclusive (the fluorophore experiment provided in the response document actually suggests that the CLR01 molecule is not able to traverse the membrane and, rather, gets stuck in the endosome). However, on balance, the manuscript is significantly improved and publication is recommended.

We thank the Reviewer for the very positive comments on our new work on the pharmacokinetics and for noting that **“the manuscript is significantly improved and publication is recommended.”** The fluorophore experiment mentioned by the reviewer, and provided in the response document, is part of further work underway on the exact mechanisms by which CLR01 crosses the cell membrane which will be published separately.